# Porcine intraepithelial lymphocytes undergo migration and produce an antiviral response following intestinal virus infection

Yuchen Li[1], Yichao Ma[1], Yuxin Jin[1], Xuebin Peng[1], Xiuyu Wang[1], Penghao Zhang[1], Peng Liu[1], Chun Liang[1] & Qian Yang [1✉]

The location of intraepithelial lymphocytes (IELs) between epithelial cells provide a first line of immune defense against enteric infection. It is assumed that IELs migrate only along the basement membrane or into the lateral intercellular space (LIS) between epithelial cells. Here, we identify a unique transepithelial migration of porcine IELs as they move to the free surface of the intestinal epithelia. The major causative agent of neonatal diarrhea in piglets, porcine epidemic diarrhea virus (PEDV), increases the number of IELs entering the LIS and free surface of the intestinal epithelia, driven by chemokine CCL2 secreted from virus-infected intestinal epithelial cells. Remarkably, only virus pre-activated IELs inhibits PEDV infection and their antiviral activity depends on the further activation by virus-infected cells. Although high levels of perforin is detected in the co-culture system, the antiviral function of activated IELs is mainly mediated by IFN-γ secretion inducing robust antiviral response in virus-infected cells. Our results uncover a unique migratory behavior of porcine IELs as well as their protective role in the defense against intestinal infection.

[1] MOE Joint International Research Laboratory of Animal Health and Food Safety, college of veterinary medicine, Nanjing Agricultural University, Weigang 1, Nanjing, Jiangsu 210095, PR China. ✉email: zxbyq@njau.edu.cn

The small intestine represents the greatest surface of the body in contact with the external environment[1]. As the important part of the immune surveillance of the intestinal mucosa, intestinal intraepithelial lymphocytes (IELs) constitute the largest heterogeneous population of long-lived resident lymphocytes in the intestinal epithelial layer[2,3]. The strategy of IELs location between adjacent epithelial cells in the lateral intercellular space (LIS) and along the basal surface, allows them to initiate timely response and provide a first line of immune defense against enteric viral infection[4,5]. Under homeostatic conditions, the IELs are highly motile and patrol the entire length of the villus and exhibit a flossing behavior. These cells actively migrate almost along the basement membrane, and occasionally exhibit transient back-and-forth movement from the basal epithelial surface into the LIS[6,7]. The surveillance behavior is altered in the presence of an invading microorganism, which generates "hotspots" of invasion for recruited IELs migrating nearby[8]. However, the molecular mechanisms regulating basal IELs migration within the epithelial compartment remain unknown[9,10]. Moreover, such strategic localization and migration also have drawn extensive research interest on how IELs promote virus resistance within the intestinal epithelia.

Represented by conventional (induced) and unconventional (natural) T-cell subsets, the intestinal IELs compartments are distinguished by their T-cell receptor (TCR) and CD8 co-receptor expression, which primarily consists of $\gamma\delta^+$ TCR CD8$\alpha\alpha^+$ and $\alpha\beta^+$ TCR CD8$\alpha\alpha^+$ cells[11]. In this context, these "activated yet resting" T cells are characterized by high expression levels of activation markers, NK-like receptors, cytotoxic T lymphocyte-related genes, and anti-inflammatory or inhibitory receptors[12]. Their biological function was previously assumed to depend on TCR activation. Studies have shown that CD3 antibodies could activate IELs, thereby inducing an antiviral response in intestinal epithelial cells (IECs) and protecting them against norovirus infection[13,14]. However, a coordinated IECs-IELs defense against enteric *Salmonella typhimurium* infection has been reported recently, in which epithelial cells microbial sensing and signaling was a requirement for IELs metabolic switch, while also supporting the maintenance of the intestinal epithelial barrier[9]. The TCR-independent activation pattern of IELs should be investigated further to elucidate their innate immune properties. Gut IELs have repeatedly been reported to exert constitutive cytotoxic activity after oral infection reovirus and rotavirus, as well as systemic lymphocytic choriomeningitis virus (LCMV) infection[15–17]. Recent studies suggested that these IELs also produce many biologically active soluble mediators, including several antimicrobial peptides and anti-infection cytokines, which collectively facilitate early protection against mucous layer invasion by intestinal pathogens[18,19]. All these IELs effector functions are stringently regulated to prevent aberrant cytolytic activity; however, the specific regulatory mechanisms involved need to be elucidated.

As a highly virulent enteropathogenic coronavirus, the porcine epidemic diarrhea virus (PEDV) has caused major economic loss to pig industries worldwide. PEDV primarily infects porcine intestinal epithelial cells (IECs) and causes superficial villous epithelial cells necrosis throughout the small intestine[20,21]. The strategic location and antiviral activity of intestinal IELs, as well as their migratory behavior, strongly argued for the further exploration of their role in resisting intestinal PEDV infection. In the present study, a distinct transepithelial migration pattern of IELs was identified in the porcine small intestinal mucosa. The co-culture models of IELs and IECs, as well as the ligated intestinal loop model were established to investigate the migration behavior of intestinal IELs. Furthermore, we further explored the molecular mechanisms underlying IELs recruitment, activation, and antiviral function in response to intestinal PEDV infection. Our results reveal the underlying protective mechanisms of porcine intestinal IELs, highlighting their crucial role in defense against intestinal virus infection.

## Results

**The IELs move to both intercellular and free surface of porcine intestinal epithelia**. Histological sections stained with hematoxylin and eosin (HE) showed the distribution of IELs in the small intestine. The IELs with a high nucleus/cytoplasm ratio were detected in the basal, intercellular, as well as the free surface of intestinal epithelia (Fig. 1a). As approximately 90% of all intestinal IELs express TCRs[2], CD3 protein staining was used to further assess the presence of IELs in the small intestinal epithelia. Consistent with HE staining, although most IELs were in the basal or intercellular surface of epithelial layers, some occasionally reached the free surface or even migrating into the intestinal lumen of the small intestine (Fig. 1b, c). The morphological appearance of IELs in the free surface of jejunal epithelia was further examined by scanning electron microscopy, which maintained their structural integrity and overall morphology (Fig. 1d). Moreover, immunofluorescence staining revealed that the collected luminal IELs were positive for cytoplasmic CD3 and preserved their proliferative activity (Fig. 1e). By comparing the distribution of the intestinal IELs in all pig growth stages, IELs were more frequently identified in small intestine tissues; the jejunum exhibited the highest IELs distribution after weaning ($p < 0.05$) (Supplementary Fig. 1 and Fig. 1f). IELs increased substantially at all intestinal sites as piglets grew, although few IELs were detected in the intestinal tissues of suckling pigs (1- or 6-day-old; Supplementary Fig. 1 and Fig. 1f). An age-dependent increase in intraepithelial and transepithelial migration of IELs was noted, which was more frequent in the jejunum and ileum ($p < 0.05$) (Supplementary Fig. 1 and 1f).

**PEDV infection induced the intra- or trans-epithelial migration of intestinal IELs**. To explore the effect of PEDV infection on intestinal IELs, we performed animal challenge experiments using 30-day-old piglets assigned to either mock-infected or virus-infected groups and orally inoculated with phosphate-buffered saline (PBS) or PEDV, respectively. Pathological examination of the intestine of PEDV-infected piglets did not show apparent pathological damage; however, some PEDV-positive cells were detected in villus epithelial cells of the jejunum (Supplementary Fig. 2). Moreover, the intestinal tissue was fixed and subjected to IELs staining. In contrast to IELs being scattered within the epithelial and lamina propria layer of small intestines of mock-infected piglets, large amounts of intercellular and free surface located IELs were detected in entire small intestines collected from piglets with PEDV infection (Fig. 2a, b). To further explore the intestinal IELs migration influenced by PEDV, we established the intestinal ligated loop model in vivo (Fig. 2c). Immunohistochemistry (IHC) analysis showed that PBS inoculation did not influence the diffuse distribution of IELs in either the jejunum or the ileum, whereas *B. subtilis* inoculation significantly promoted the entry of IELs into the intestinal epithelial layer; some IELs moved to the free surface of the intestinal epithelia. PEDV inoculation significantly promoted the entry of IELs into the LIS of the small intestinal epithelia, and the number of epithelial layers in which IELs were located increased markedly with the virus inoculation time (Fig. 2d, e). Transepithelial migration of intestinal IELs was also identified in the jejunum and ileum of piglets injected with PEDV for only 1 h (Fig. 2d, e). The jejunum section was also subject to proliferating cell nuclear antigen (PCNA) staining for detecting the proliferation activity of intestinal IELs. PEDV inoculation significantly stimulated the

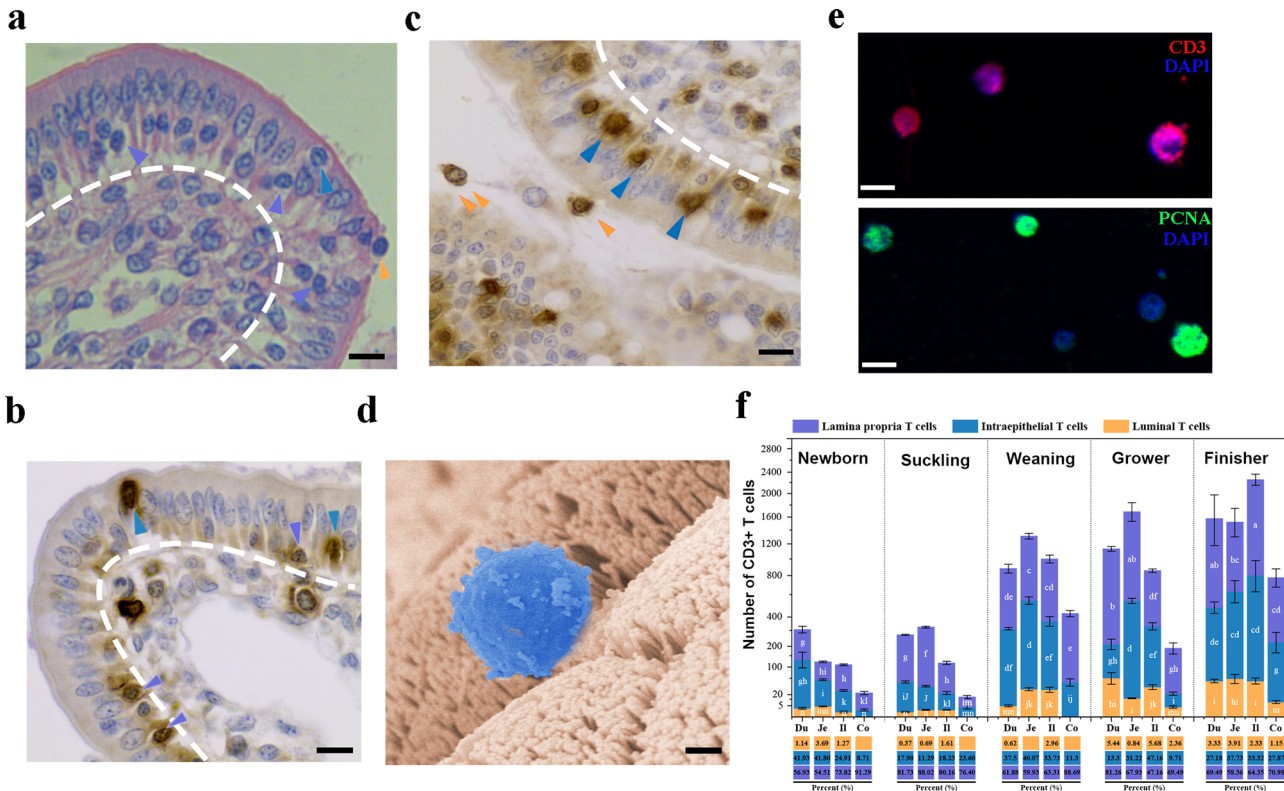

**Fig. 1 The distribution of porcine intestinal intraepithelial lymphocytes (IELs). a** Hematoxylin and eosin (HE) staining showing the distribution of IELs in the basal surface (white arrowheads), lateral intercellular space (LIS) (black arrowheads), or free-surface (red arrowheads) of epithelial layer in jejunum tissues (45-day-old piglets). Scale bars, 10 μm. **b** Representative images of the CD3-positive IELs in the basal (white arrowheads) or intraepithelial of jejunal epithelia (black arrowheads) (45-day-old piglets). Scale bars, 10 μm. **c** Representative images of the CD3-positive IELs located in LIS (black arrowheads) or free-surface (red arrowheads) of jejunal epithelia (45-day-old piglets). Scale bars, 10 μm. **d** Representative SEM image showed the luminally located IELs (blue) in jejunum tissue (45-day-old piglets). Scale bars, 5 μm. **e** IELs were isolated from ileal contents and stained for CD3 and PCNA protein. The scale bar represents 10 μm. **f** Quantitative analysis of IELs within various intestinal segments collected from pigs ($n = 3$) in different growth stages. The numbers of CD3-positive cells in different sites of the intestinal epithelial layer were counted in 20 random intestinal villi. The frequency of total T cells in each area (%) was presented in the bar chart. All data shown are the mean results ± SD from three independent experiments. Statistical significance was shown using one-way ANOVA. NS: no significance; *$P < 0.05$, **$P < 0.01$. The differences are indicated by different letters. Letters above the graphs indicate statistical significance in which treatments with a letter in common are not significantly different from each other.

proliferation of intestinal IELs with PCNA staining, particularly those moving to the free end of the intestinal epithelia (Fig. 2f).

**PEDV infection promoted the transepithelial migration of intestinal IELs in a co-culture system**. To determine the behavior change of intestinal IELs influenced by PEDV infection, the intestinal IELs were further isolated to establish an IELs/IPEC-J2 coculture system The proportion of the isolated IELs was identified by staining with specific phenotype antibodies. FACS analysis showed that more than 70% of cells expressed a CD3 receptor, in which 48.5% of the cells were also positive for anti-pig γδ T antibody (Fig. 3a). The IELs co-culture did not modify the morphological characteristics of tight junctions in IPEC-J2 or influence the co-culture system's transepithelial electrical resistance (TEER) (Supplementary Fig. 3). Thereafter, the transwell insert of IELs/IPEC-J2 coculture system was placed on PEDV-infected Marc-145 cells for a further 12 h coculture (Fig. 3b). The fluorescence-labeled IELs were identified in z-orthogonal views using confocal laser scanning microscopy (CLSM). Considering that IPEC-J2 was insusceptible to infection with PEDV, the virus released in the basal supernatant did not influence the integrity of the epithelial barrier in the co-culture system. However, compared with the inactivated virus treatment, live PEDV inoculation resulted in the migration of more IELs across the intestinal epithelia (Fig. 3c, d), and into the lower chamber of the transwell

(Fig. 3c, e). Therefore, PEDV induced the migration of IELs into the epithelial layer or the free surface of the small intestine, and enhanced the proliferative activity of intestinal IELs. Transmission electron microscopy (TEM) revealed the cellular composition of the IELs/IPEC-J2 coculture system (Fig. 3f). In response to PEDV infection, the elongated cell body of IELs were observed, which migrate into the filter or the LIS of IECs. Compared with the IELs from the mock-infected group (WI PEDV treated), the variation in morphological characteristics, such as irregular shapes and increased intracellular lipid droplets (white asterisk), were observed in the IELs from PEDV infected group.

**CCL2 secretion from PEDV-infected cells modulated IELs migration**. Relevant chemokines strictly regulated the migration of intestinal IELs under physiological conditions. To better understand the molecular cues required for the intraepithelial and transepithelial migration of intestinal IELs, we summarized and analyzed the transcript profile of chemokines (associated with immune cell recruitment) in PEDV-infected cells (PRJNA679356) or intestinal mucosa[22,23]. A Sankey diagram showed that the upregulated chemokines including, chemokine (C-C motif) ligand 2 (CCL2), CCL5, chemokine (C-X-C motif) ligand 9 (CXCL9) and CXCL10 were identified in at least two datasets, in which CCL2 was found to be upregulated in all datasets (Fig. 4a). Based on these results, we assessed the expression of screened

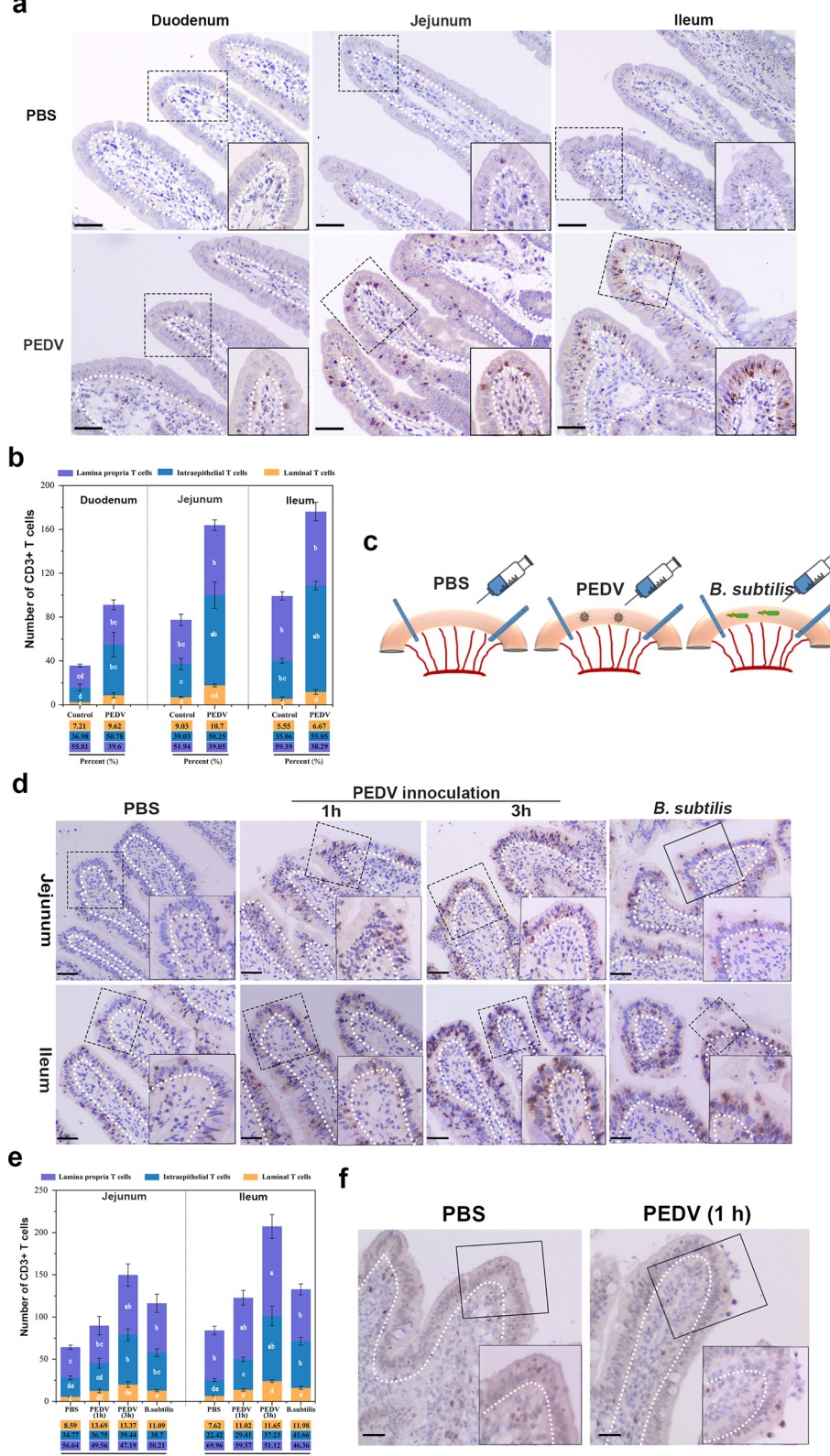

chemokines associated with PEDV infection. CCL2 expression was detected in jejunum sampled from PEDV-infected piglets, and in the terminal jejunal ligated loops inoculated with PEDV. Immunofluorescence analysis (IFA) of cryosections showed that PEDV significantly promoted CCL2 expression and accumulation in the apical of jejunal epithelia (Fig. 4b). Meanwhile, the secretion of CCL2 was also upregulated after in vitro PEDV inoculation, which showed no change in whole inactivated virus (WIV)-treated Marc-145 cells (Fig. 4c).

To further verify the role of CCL2 in intestinal IELs trafficking, recombinant porcine CCL2 or CCL5 protein were added to the medium of the basolateral side in IELs/IPEC-J2 coculture system (Fig. 4d). A CCL2 concentration of 50 ng/mL induced intestinal IELs transepithelial migration, whereas CCL5 treatment had no

**Fig. 2 Influence of porcine epidemic diarrhea virus (PEDV) infection on porcine intestinal intraepithelial lymphocytes (IELs). a** Pigs ($n = 3$) from mock or PEDV challenge groups were euthanized at 48 hpi, and small intestinal tissues were fixed and subjected to IHC analysis. Representative image of IELs distribution in the small intestine of PBS-inoculated or PEDV-inoculated weaned piglets (one-month-old). Scale bars, 20 μm. **b** Quantitative analysis of IELs showed in (**a**) was performed, in which 20 random intestinal villi were counted. The frequency of total T cells in each area (%) were also presented. **c** Schematic of the experimental setting of the intestinal ligated loop model. Anesthetized piglets (one-month-old, $n = 3$) were subjected to intestinal ligation followed by injection with PBS, PEDV, and *Bacillus subtilis*, and were euthanized at 3-hours post-treatment. **d** Immunohistochemistry (IHC) analysis demonstrated the distribution pattern of IELs influenced by PEDV or *B. subtilis* in the ligated loop of the terminal jejunum. Scale bars, 20 μm. **e** Quantification of IELs (**d**) in 20 random intestinal villi. The frequency of total T cells in each area (%) was also calculated. **f** The proliferative activity of intraepithelial and transepithelial IELs in the ligated loop was detected via proliferating cell nuclear antigen (PCNA) staining. The scale bar represents 50 μm. All data shown are the mean results ± SD from three independent experiments. Statistical significance was obtained using one-way ANOVA. NS no significance, *$P < 0.05$, **$P < 0.01$. The differences are indicated by different letters. Letters above the graphs indicate statistical significance in which treatments with a letter in common are not significantly different from each other.

stimulating effect at concentrations of 5–100 ng/mL (Fig. 4e, f and Supplementary Fig. 4). However, pretreatment of the intestinal IELs with a CCL2 receptor inhibitor [CCR2 antagonist Teijin Compound 1 (TC1)] significantly counteracted the function of CCL2 in intestinal IELs recruitment (Fig. 4e, f). The proliferation assay results showed that neither of the CCL2, CCL5, and TC1 pretreatments significantly influenced the viability of intestinal IELs or the integrity of the epithelial barrier in the co-culture system (Supplementary Fig. 5). TC1 pretreatment significantly inhibited the migratory behavior of intestinal IELs induced by PEDV in the co-culture system (Fig. 4g, h).

**Virus-activated IELs protected epithelial cells from PEDV infection.** The TCR-dependent IELs activation protected cells against norovirus infection (*14*). To explore whether intestinal IELs have a protective effect on PEDV infection, PEDV-infected epithelial cells were cultured with intestinal IELs (CD3 antibody-stimulated) (Fig. 5a). However, the viral protein and mRNA expression levels and the release of infectious virion were not affected by intestinal IELs (Fig. 5b–d). Our previous studies showed that intestinal IELs were recruited into the LIS and the free surface of the intestinal epithelia in response to PEDV infection. Therefore, we supposed that this migration pattern of intestinal IELs may facilitate direct contact with viral antigen and be activated simultaneously. To test this, we further assessed the antiviral activity of intestinal IELs stimulated by either the whole inactivated (WI) PEDV or the PEDV spike (S) protein. Although S protein stimulation did not confer antiviral effects of IELs, the co-culture significantly inhibited PEDV infection in epithelial cells with virus-activated intestinal IELs, which yielded a nearly 80% reduction in progeny virus and 50% reduction in viral protein and mRNA expression (Fig. 5b–d). To further explore the potential mechanisms involved in the protective function of porcine intestinal IELs, we established a co-culture mode in vivo, in which PEDV-infected epithelial cells were cultured with pre-activated intestinal IELs in two forms, contact or noncontact (Fig. 5e). The antiviral effect of IELs was evaluated by detecting the release of infectious virus particles, and the expression levels of intracellular viral RNA and protein (Fig. 5f–h). Pre-activated IELs exhibited a similar and significantly inhibitory effect on PEDV infection in both culture models, which revealed that the antiviral activity of porcine intestinal IELs might be induced by the TCR-independent signaling of virus-infected cells. Meanwhile, we also suggested that intestinal porcine IELs could exert antiviral activity via the secretion of biologically-active mediators.

**In vivo pre-activation endowed IELs with potent antiviral activity against PEDV infection.** To further verify whether the antiviral activity of IELs required pre-activation by viral antigen, 30-day-old piglets were orally inoculated either with 1 mL WI

PEDV ($10^7$ PFU/mL) or PBS for 4 h, and the same volume of PBS was used as a negative control (Fig. 6a). Fluorescence-activated cell sorting (FACS) of IELs isolated from epithelial layers showed that the proportion of CD3$^+$T cells increased significantly after PEDV inoculation (Fig. 6b). IHC results revealed that PEDV inoculation significantly promoted the intraepithelial and trans-epithelial migration of intestinal IELs in the small intestinal mucosa (Fig. 6c, d). The intestinal IELs isolated from mock-infected piglets exerted an antiviral effect dependent on viral pre-activation. The IELs isolated from PEDV-inoculated piglets significantly suppressed viral infection without preactivation, which showed similar antiviral efficiency with preactivated intestinal IELs (Fig. 6e, f). Moreover, the antiviral function of T cells in the mesenteric lymph node (MLN) was also assessed in each group. Resembling intestinal IELs, the antiviral function of mesenteric lymph node (MLN) T cells in mock-infected piglets was identified only upon PEDV pretreatment, while MLN-T cells isolated from PEDV-inoculated piglets exhibited an antiviral effect without pre-activation. In contrast, the antiviral activity of intestinal IELs was significantly more potent than that of MLN T cells (Fig. 6e, f).

**Cytotoxicity of pre-activated intestinal IELs eliminated virus-infected cells.** CD8α$^+$ γδ and CD8α$^+$ αβ T cells account for the main subgroup of porcine intestinal IELs (Fig. 3a), which raises concern regarding their cytotoxic effect, endowing them with the capacity to lyse infected or aberrant cells. Therefore, PEDV-infected, or whole inactivated PEDV-treated epithelial cells were co-cultured with preactivated IELs, respectively (Fig. 7a). The apoptosis of epithelial cells was determined via TUNEL staining, and PEDV-infected epithelial cells were used as the negative control. As shown in Fig. 7b, the apoptosis observed in PEDV-infected epithelial cells was rare, despite a large number of viral antigens detected. The pre-activated IELs co-culture induced evident apoptosis in virus-infected epithelial cells, and viral replication was significantly inhibited. However, no apparent apoptosis was detected in the inactivated virus-treated epithelial cells, even those co-cultured with pre-activated IELs (Fig. 7b, c). Furthermore, the lactate dehydrogenase (LDH) release assay was used to evaluate the specific cytotoxic of IELs on epithelial cells. Consistent with the results of apoptosis detection, the cytotoxicity of preactivated IELs was significantly enhanced when they were co-cultured with PEDV-infected epithelial cells (Fig. 7d). The cytotoxic activity of IELs can be inferred from their potent expression of granzymes and perforins. As shown in Fig. 7e, perforin expression was highly elevated in IELs when they were co-cultured with PEDV-infected epithelial cells; however, non-significant changes in granzyme B expression were detected (Fig. 7f). According to the above result, we further explored the antiviral function of IELs after blocking perforin and granzyme activity. Concanamycin A (CMA) and 3, 4-dichloroisocoumarin (DCI) were used to block perforin and granzyme activity,

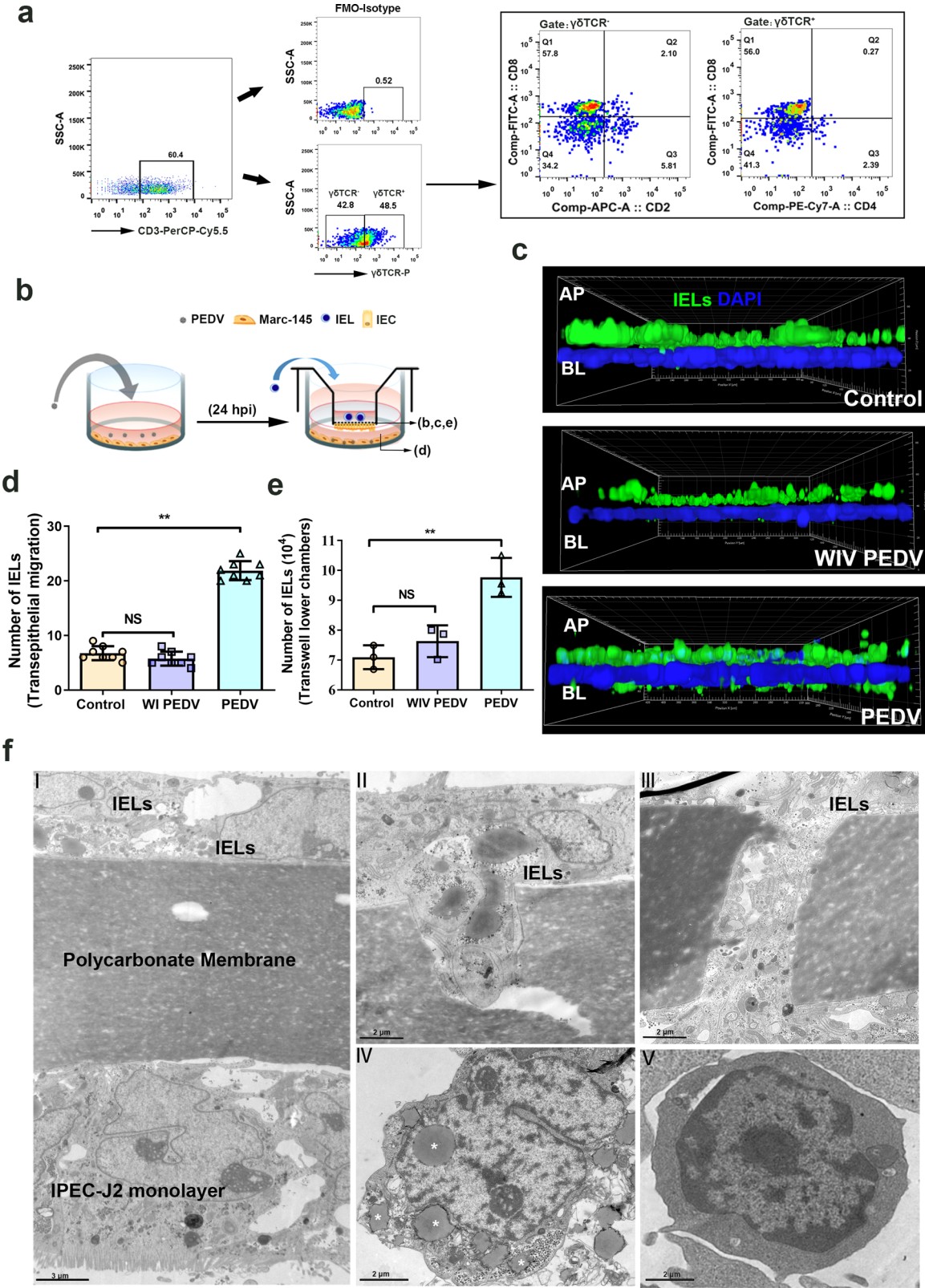

respectively[24,25]. However, the antiviral activity of IELs were not influenced by treating them with different concentrations of granzyme or perforin inhibitor (Fig. 7g). Similarly, the inhibitor treatment also did not reverse the apoptosis of virus-infected epithelial cells induced by IELs (Fig. 7h). The IELs viability of each experimental group was not influenced by the inhibitors used (Supplementary Fig. 6).

**Pre-activated intestinal IELs induced antiviral responses by producing interferon (IFN)-γ.** IELs have also been also reported to participate in the innate immune defense of intestinal mucosa through IFN secretion. To explore the role of IFN in viral resistance of porcine intestinal IELs, the following groups were set: PEDV-infected epithelial cells, IELs (pre-activated/unstimulated) co-cultured with epithelial cells (treated with whole inactivated

**Fig. 3 Porcine epidemic diarrhea virus (PEDV) infection promotes the transepithelial migration of intestinal intraepithelial lymphocytes (IELs) in an in vitro co-culture system. a** Flow cytometry analysis revealed population and subpopulations of isolated porcine intestinal IELs. **b** Schematic of the experimental setting to study the migration of IELs in the co-culture system. Mac-145 cells (70% confluent) were inoculated with PEDV at a multiplicity of infection (MOI) of 0.1 for 24 h at 37 °C. Next, the transwell insert containing the IELs/IPEC-J2 co-culture system was cultured with PEDV-infected Marc 145 cells for 2 h. **c** The migration of IELs in the co-culture system was determined using immunofluorescence. The filters in the coculture system were processed and viewed using confocal laser scanning microscopy (CLSM; AP to BL). CFSE labeled IELs (green) were detected in the apical or basal side of the monolayer intestinal epithelial cells (DAPI, blue). **d** Quantification of basolateral IELs from fluorescence images in (c); each value was calculated from the averaged values of three individual filters. **e** The number of CFSE-labeled IELs in lower chambers of co-culture system were counted by FACS. **f** The filters from the co-culture system were processed for TEM. The representative image shows the cellular composition of the co-culture system (panel I), and the migrating IELs (panels II and III). Moreover, the IELs in apical side (mock-infected group; panel IV) or basal surface (PEDV infected group; panel V) of IPEC-J2 in the co-culture system were also collected separately and sent for TEM observation. The scale bar represents 3 μm (I); and 2 μm (II to V). All data shown are the mean results ± SD from three independent experiments. Statistical significance was obtained using one-way ANOVA. NS no significance, *$P < 0.05$, **$P < 0.01$.

PEDV), and (pre-activated/unstimulated) IELs co-cultured with epithelial cells (infected with PEDV) (Fig. 8a). After co-culturing, the culture supernatant was collected for detecting IFN secretion. The ELISA results showed that there was no significant change in the secretion of types I (IFN-α and IFN-β) or III IFN (IFN-γ) with IELs (pre-activated/unstimulated)-co-cultured epithelial cells (PEDV-infected). However, the secretion of type II IFN (IFN-γ) increased significantly when pre-activated IELs were co-cultured with virus-infected epithelial cells, although the secretion of type I IFN (IFN-α and IFN-β) did not change markedly (Fig. 8b–d). Significantly upregulated transcription of IFN-γ in activated IELs was also detected in the co-culture with virus-infected cells (Fig. 8e). The IFN-γ secreted by IELs upregulated the expression of IFN-stimulated genes (ISGs) and induced a potent antiviral effect in epithelial cells. Therefore, the activation status of STAT1, as well as the transcription of ISGs in epithelial cells were further detected. Although the level of STAT1 phosphorylation was significantly inhibited by PEDV infection, a significantly elevated level of p-STAT1 was observed in epithelial cells co-cultured with virus-activated IELs (Fig. 8f). Simultaneous, activated IELs co-culture also significantly promoted the transcription of ISGs (MX2, IFIT3, OAS, viperin, IFITM1, and ISG15) in virus-infected epithelial cells, whereas PEDV infection alone did not influence the transcription of ISGs in epithelial cells (Fig. 8g). Although the upregulation of IFIT3, OSA, and IFTMI was also identified in PEDV-infected epithelial cells (non-activated IELs co-culture), their expression levels were significantly lower than those detected in epithelial cells co-cultured with activated IELs. Additionally, neither the activated nor the inactivated IELs stimulated the transcription of ISGs in epithelial cells treated with whole inactivated PEDV. To further verify whether porcine intestinal IELs resist PEDV infection only by secreting IFN-γ, the antibody against porcine IFN-γ was added to the co-culture system for neutralizing IFN-γ produced by IELs. After the IFN-γ antibody treatment, the antiviral effect of activated IELs was reversed in a dose-dependent manner (Fig. 8h, i), particularly at high concentrations (25 μg). Consistent with the result described above, IFN-γ antibody treatment also prevented IELs-induced apoptosis in a dose-dependent manner (Supplementary Fig. 7).

## Discussion

The intestinal IELs composition varies with species; the human small intestine is dominated by TCRαβ+ IELs, whereas the mouse has a somewhat equal distribution of TCRγδ+ and TCRαβ+ IELs[26,27]. As in humans, the majority of IELs in the small intestine of weaned pigs are αβ TCR CD8α+ cells (~80%), the percentage of which increases over time across all intestinal sites[28]. Our study further revealed the distribution characteristic of intestinal IELs in all pig growth stages

(newborn, suckling, weaning, grower, and finisher). Similar age- and location-dependent characteristics of intestinal IELs were identified; however, the deficiency or few IELs in the intestinal mucosa of newborn and suckling piglets, makes them vulnerable to intestinal infection[29]. Significantly, the transepithelial migration of intestinal IELs was discovered, which even into the lumen with small amounts. Compared to intestinal IELs in rodents, which only move transiently into the LIS between two adjacent enterocytes[30,31], porcine IELs could migrate to the free surface of the intestinal epithelia. Although the specific functions and mechanisms involved remain unverified, the unique migration pattern may be conducive to the direct contact of IELs with and identification of intestinal pathogens. Additionally, we found that the intercellular movement of IELs did not destroy the integrity of the tight junctions of the epithelium. IELs appear to traffic primarily between adjacent epithelial cells through the transient opening of tight junctions, similar to what has been reported for transepithelial dendrites formation of intestinal dendritic cells (Supplementary Fig. 8).

Enteric infections resulted in fast and well-defined behavioral changes in intestinal IELs, including reduced vertical displacement, altered villus positioning, and heightened interepithelial flossing movement. This infection-induced flossing movement preferentially occurred in specific hotspot areas after *Toxoplasma* or *Salmonella* infections[32]. Such strategic localization and migration within the intestinal mucosa may facilitate the interaction of IELs with enterocytes to prevent pathogenic incursion. In our study, PEDV oral infection significantly promoted the movement of intestinal IELs into the LIS of IECs. Furthermore, in an intestinal ligated loop model in vivo, the intraepithelial and transepithelial migration of IELs was significantly triggered by 1 h of virus treatment. The migration movement of IELs is tightly regulated, as they reside in the intestinal mucosa by expressing chemokine receptor CCR9 and integrin α4β7 and interact with the ligand CCL25 and E-cadherin in IECs[33]. Additionally, the intestinal IELs recruitment in inflamed gut tissues is preferentially guided by CXCR3 and its ligands CXCL10[34]. In our study, the chemokine CCL2 played an important role in PEDV-infected cells and induced the migration of intestinal IELs, whereas pretreatment with CCR2 inhibitor ligands partly reversed this recruitment effect. The accumulation of CCL2 in the apical side of IECs might be highly beneficial to the recruitment of intestinal IELs into the epithelial layer or the apical side along the direction of the concentration gradient. The CCL2 has been reported to exhibit chemotactic activity in regulating monocytes and basophils' activation, migration, and infiltration[35]. Our study revealed the alternative function of epithelial-derived CCL2 in stimulating intestinal IELs migration behavior in response to intestinal infection.

Previous studies have shown that intestinal IELs profoundly limited norovirus infection in a TCR-dependent manner[14].

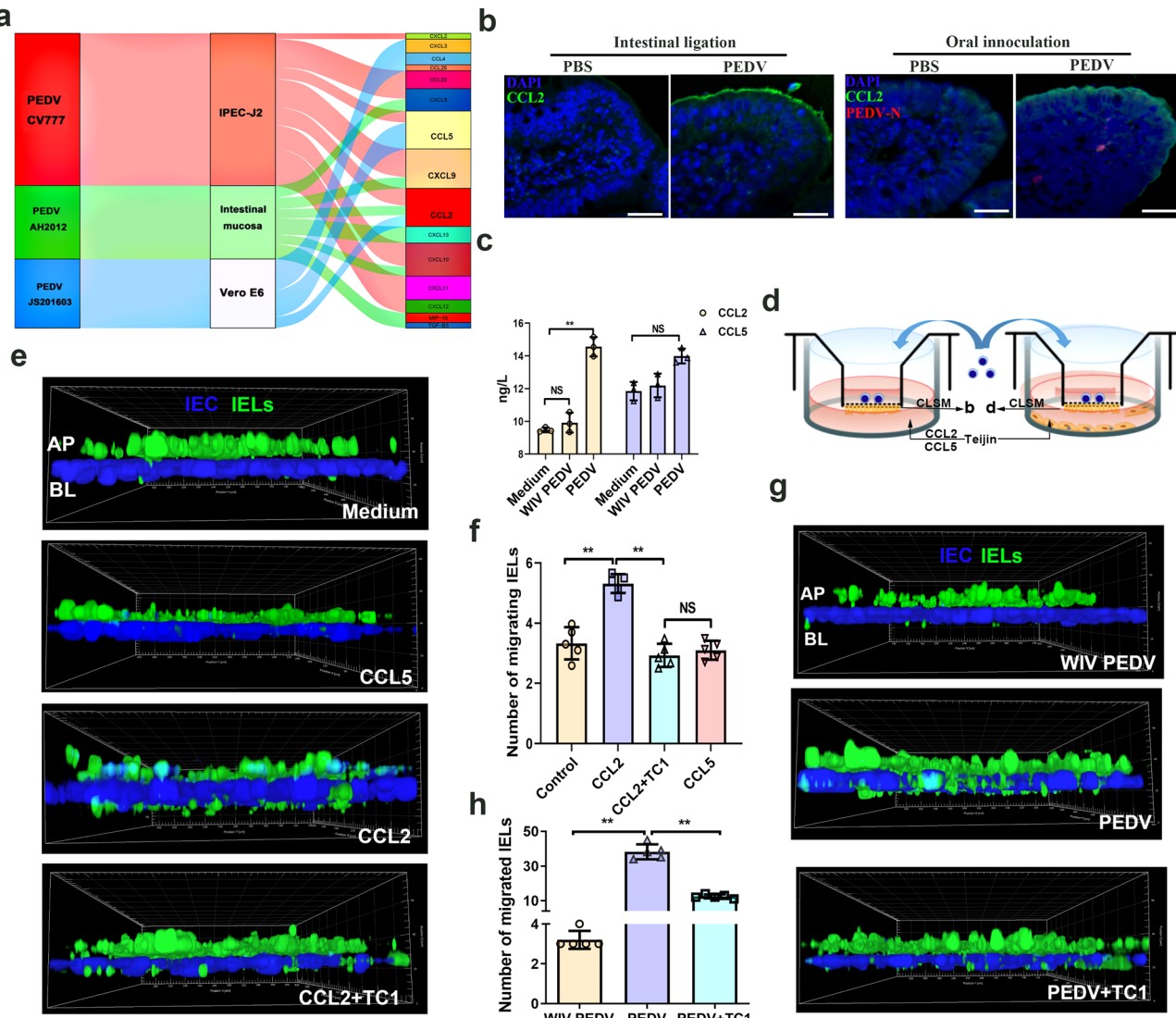

**Fig. 4 Role of CCL2 in the intraepithelial and transepithelial migration of intestinal intraepithelial lymphocytes (IELs) induced by porcine epidemic diarrhea virus (PEDV) infection. a** Sankey diagram depicts the chemokine secretion in the host cells and the intestinal mucosa infected with variant and classical PEDV strains. **b** The protein expression of CCL2 in the ligated terminal jejunum injected with PEDV and in the jejunum from piglets orally inoculated with PEDV was detected via immunofluorescence analysis. CCL2 and PEDV were immunolabeled with anti-CCL2 mAb (green) and anti-PEDV polyclonal antibody (Red), respectively; the cell nuclei were stained with DAPI (blue). Scale bars, 50 μm. **c** The protein expression of CCL2 and CCL5 in the medium of Marc-145 cells was detected using ELISA kits. **d** Schematic of experimental setting used to study the migration of IELs influenced by chemokines induced by PEDV infection. **e** Chemokines CCL2, CCL5, and the CCL2 inhibitor TC1 were added into the medium of the basolateral side. Filters from the co-culture system were determined via CLSM, and the intraepithelial and transepithelial (DAPI, blue) migration of IELs (CFSE, green) in response to different treatments was shown by a three-dimensional (3D) rendering of representative fields. **f** Quantitative analysis of transepithelial IELs was performed. The number of transepithelial IELs was counted from five random fields of view at a unit area (0.078 mm$^2$) for each of the three individual filters. **g** The IELs/ IPEC-J2 co-culture model was pretreated with an CCL2 inhibitor TC1 for 2 h (DMSO as a negative control) by upper compartment inoculation, followed by a culture with PEDV-infected or whole inactivated PEDV-treated Marc-145 cells. The intraepithelial and transepithelial (DAPI, blue) migration of IELs (CFSE, green) was detected via CLSM. **h** Quantitative analysis of transepithelial IELs in each experimental group. All data are the mean ± SD, and comparisons were performed using one-way ANOVA. *$P < 0.05$, **$P < 0.01$. The results are from at least three different experiments.

However, in response to *S. typhimurium* infection, the protective role of intestinal IELs was shown to be independent of TCR signaling, which was mediated via IELs/epithelial crosstalk downstream of epithelial MyD88 signaling[9]. In the present study, we found that only whole virus-activated IELs inhibited PEDV infection in epithelial cells, which exert antiviral activity depending on the reactivated signaling from virus-infected epithelial cells. Identically, oral inoculation of whole inactivated (WI) PEDV endowed the antiviral activity of porcine IELs independent of WI virus pre-activation. WI PEDV is hard to

direct contact with the antigen-presenting cells, such as macrophages and dendritic cells, as well as other innate immune cells in the gut lamina propria, due to the loss of replication in the intestinal epithelia[36]. Therefore, we supposed that intestinal IELs may get into a preactivation state by directly contacting viral antigens through their unique transepithelial migration pattern. Subsequently, the preactivated IELs could be recruited to nearby infected IECs, where they can be further activated and mediate antiviral resistance. Although we speculated that the transepithelial migration of porcine intestinal IELs may be conducive to

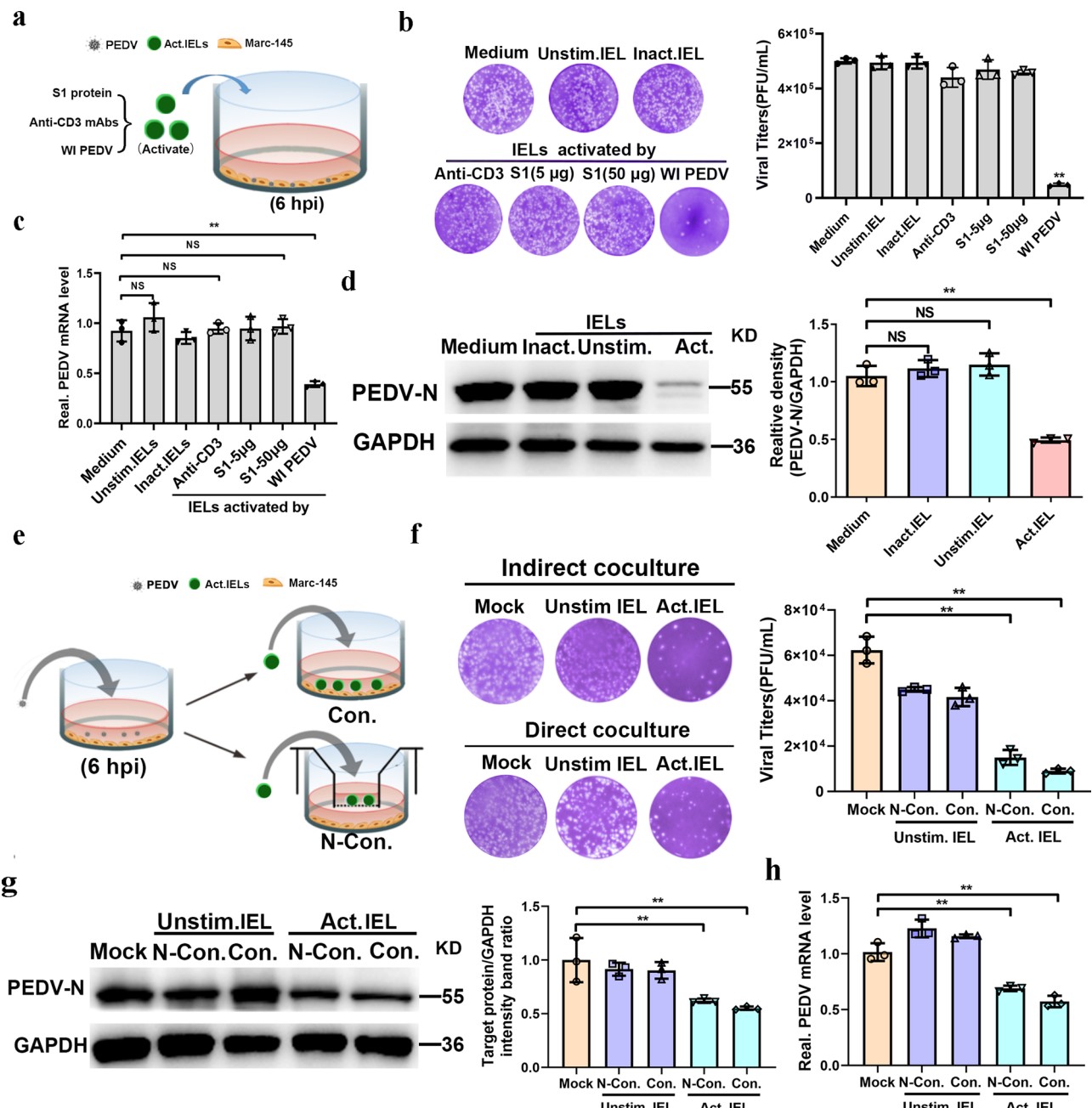

**Fig. 5 Preactivated intestinal intraepithelial lymphocytes (IELs) protect epithelial cells from porcine epidemic diarrhea virus (PEDV) infection in a TCR-independent manner.** Experimental setting to study the antiviral effect of intestinal IELs. **a** The scheme depicts that intestinal IELs were pretreated with CD3 antibody, whole inactivated PEDV, and the PEDV spike protein; thereafter, they were co-cultured with PEDV-infected epithelial cells for 24 h. **b** The viral titers in the culture supernatant were detected via a plaque assay and summarized in a histogram. **c** The intracellular viral RNA levels were quantitated via RT-qPCR. **d** The expression of PEDV-N protein in epithelial cells (Marc-145) was detected using western blot. **e** The model used to study the protective role of intestinal IELs against PEDV infection. The scheme depicts the whole inactivated PEDV pre-activated or unstimulated IELs that were co-cultured with epithelial cells (Marc-145) using two methods (contact and noncontact co-culture). **f** Viral titers in the supernatant of the coculture system were measured via a plaque assay after a co-culture of 24 h. Meanwhile, the expression of the viral protein (**g**) and mRNA e (**h**) in epithelial cells (Marc-145) was also evaluated. All data are the mean ± SD, and comparisons were performed using one-way ANOVA. *$P < 0.05$, **$P < 0.01$. The results are from at least three different experiments.

their antigen recognition and activation promoted by an intestinal infection. The underlying link between the migratory behavior of IELs and their antiviral function remain undefined and require further experimental demonstrations. Furthermore, our studies also found that the recombinant spike protein could not stimulate IELs, which implies that the virus recognition of IELs required complete and complex antigen epitopes[37].

In the present study, CD3+CD4-CD8+ subsets of IELs acted as the main undertaker of the antiviral function of intestinal IELs. Considering the CD8+ T cell-mediated protective effect feature, we further detected the cytotoxicity activity of porcine IELs. Our results suggested that the antiviral function of IELs was independent of cytotoxicity, although the perforin secretion of activated IELs was highly elevated in the co-culturing system.

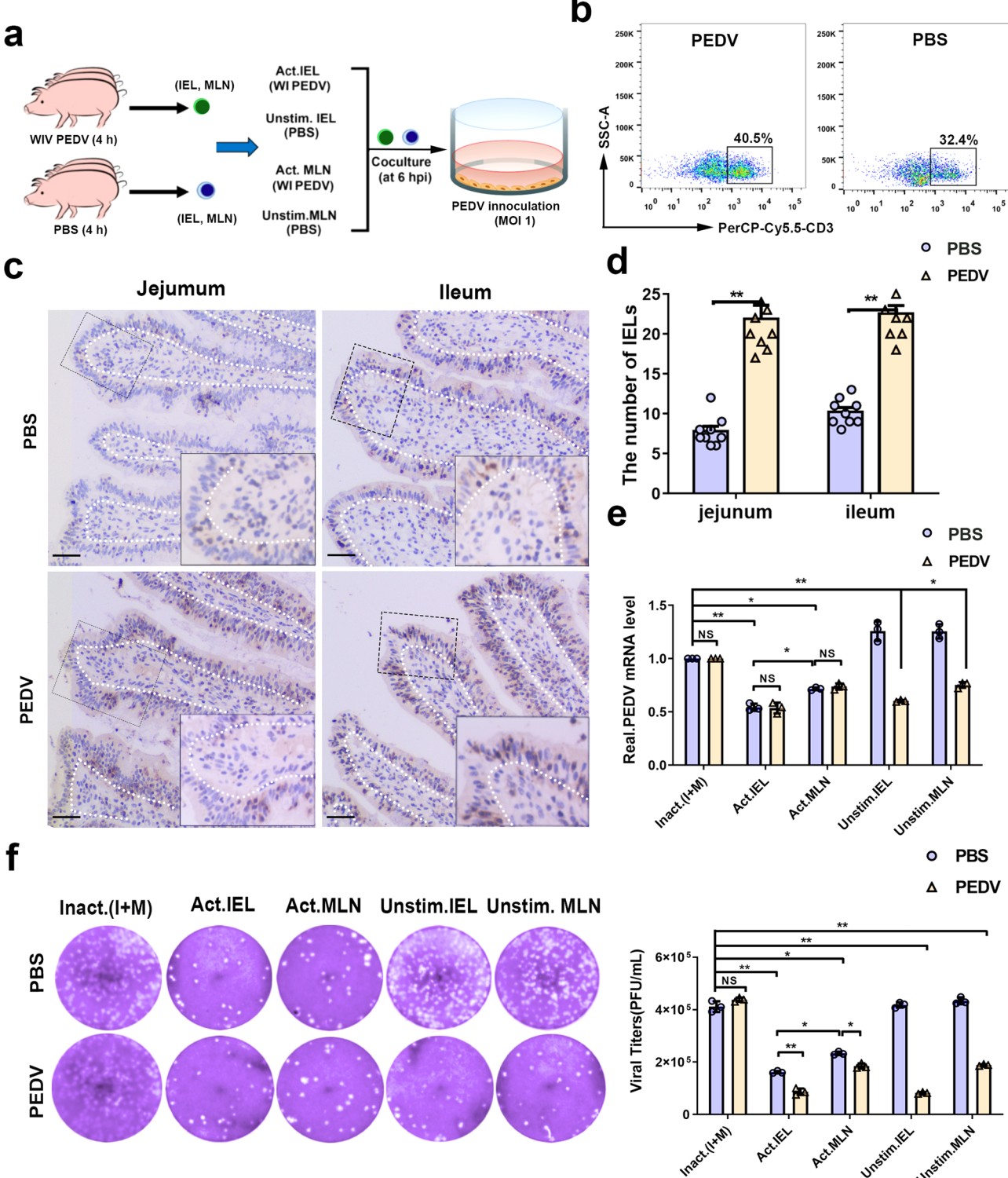

**Fig. 6 Oral inoculation with porcine epidemic diarrhea virus (PEDV) pre-activated intestinal intraepithelial lymphocytes (IELs) and increased their antiviral activity against virus infection. a** For in vivo IELs activation experiment, piglets were sacrificed 4 h after PEDV inoculation, $n = 3$ piglets per group. The scheme depicts that intestinal IELs isolated from piglets orally inoculated with PEDV (PBS inoculation was used as a negative control) were pretreated with whole inactivated PEDV for pre-activation (DMEM medium was used as a negative control) and then co-cultured with PEDV-infected epithelial cells for 24 h. **b** The percentage of CD3-positive IELs in the jejunum of piglets were analyzed using fluorescence-activated cell sorting (FACS). **c** Immunohistochemistry (IHC) analysis displayed the distribution pattern of IELs influenced by PEDV inoculation. Scale bars, 10 μm. **d** The numbers of IELs shown in (**c**) were quantified and displayed on a histogram. **e** The PEDV RNA expression in epithelial cells from the co-culture system in (**a**) was quantitated using RT-qPCR. **f** Moreover, the infectious viral particles in the culture medium of the coculture system in (**a**) were detected by plaque formation. The histogram summarizes the plaque assay results. All data are the mean ± SD, and comparisons were performed using one-way ANOVA. *$P < 0.05$, **$P < 0.01$. The results are from at least three different experiments.

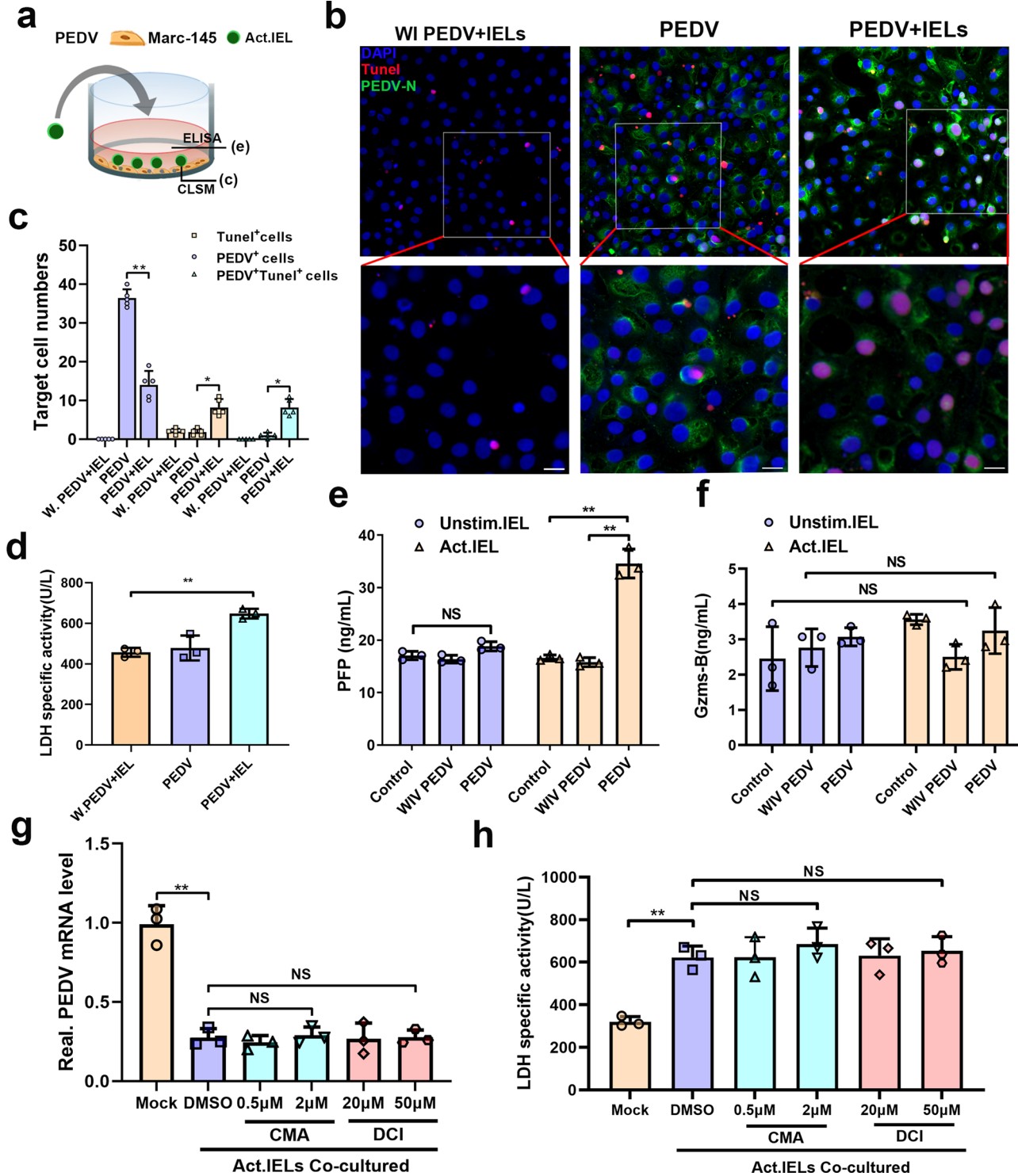

**Fig. 7 The antiviral activity of IELs was independent from their cytotoxic effects. a** The design for detecting the cytotoxicity of IELs. The virus pre-activated IELs were co-cultured with porcine epidemic diarrhea virus (PEDV)-infected epithelial cells for 24 h. **b** TUNEL staining was used to determine the apoptosis rate of PEDV-infected or whole inactivated PEDV-treated epithelial cells. **c** The number of TUNEL⁺, PEDV⁺, and TUNEL⁺PEDV⁺ cells in each group was calculated and shown in a bar graph, which was produced from five random fields of view for each of three individual sections. **d** The released lactate dehydrogenase (LDH) activity assay was also used to evaluate the cytotoxicity of IELs in different groups. **e, f** Levels of perforin (**e**) and granzyme B (**f**) release in the culture supernatants of different groups were measured via ELISA. **g, h** The pre-reactivated IELs were treated with CMA at two concentrations (1 and 0.5 µM) or DCI at two concentrations (20 and 50 µM) for 3 h, and their antiviral activity were further detected by co-culturing with virus-infected epithelial cells. Both of the inhibitors were also maintained for the whole duration of the co-culturing process. The intracellular viral RNA expression (**g**) and extracellular LDH activity (**h**) in each experimental group were determined. All data are the mean ± SD and comparisons were performed using one-way ANOVA. *$P < 0.05$, **$P < 0.01$. The results are from at least three different experiments.

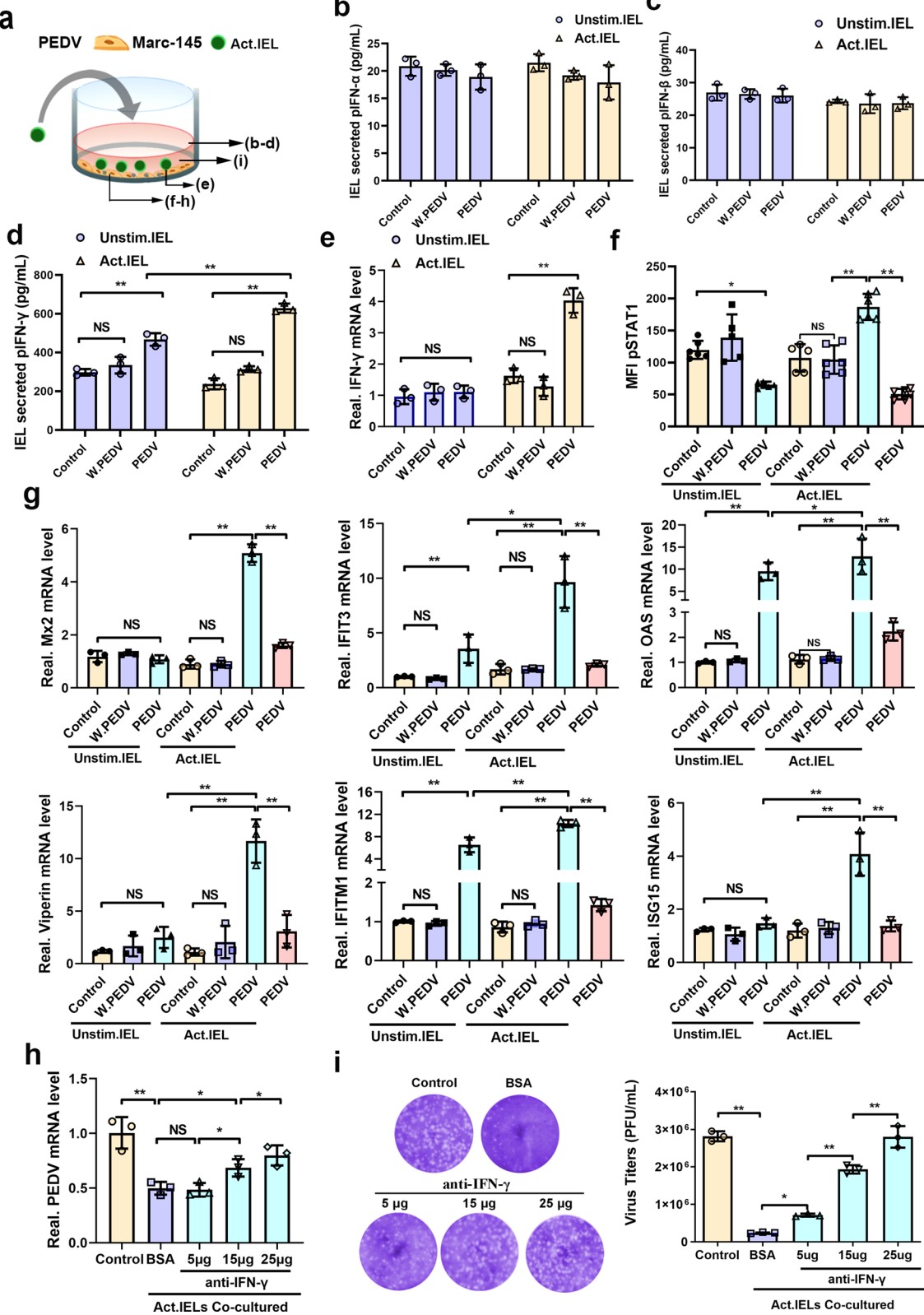

Activated intestinal IELs can also produce large amounts of biologically active soluble mediators to contribute to anti-pathogen responses[19,38]. In our ex vivo co-culture model, the activated porcine intestinal IELs exerted antiviral effects by IFN-γ production, which upregulate many ISGs suppressed by PEDV infection[39,40], thereby enhancing the epithelial antiviral response. Furthermore, it is worth noting that the upregulation of IFN-γ

was only detected in the reactivated intestinal IELs, which experienced the stimulation from viral antigen and virus-infected host cells. However, the antibody for INF-γ significantly reversed the antiviral effect of intestinal IELs. Particularly in the high-dose group. 25 μg/ml IFN-γ antibody almost completely blocked the antiviral activity of intestinal IELs. Consistently, IELs-mediated apoptosis of PEDV infected epithelial cells was also significantly

**Fig. 8 Pre-activated intestinal intraepithelial lymphocytes (IELs) produced IFN-γ and induced antiviral pathways in virus-infected epithelial cells.**
**a** The scheme depicts that mock or whole inactivated virus (WIV) porcine epidemic diarrhea virus (PEDV) pre-activated IELs were co-cultured with PEDV-infected epithelial cells for 24 h. **b–d** The production of type I (**b, c**), II (**d**), and III IFNs (**e**) by IELs in the co-culture system was detected using ELISA kits.
**e** The mRNA expression of IFN-γ in IELs was also measured via RT-qPCR. **f** Phosphorylation of STAT1 in the epithelial cells was analyzed by flow cytometry.
**g** The transcription of a set of IFN stimulated genes (ISGs) in PEDV-infected epithelial cells were detected via RT-qPCR. **h** A similar experiment was performed as in (**a**), but blocking antibodies against IFN-γ were added to the medium with a certain concentration gradient. **i** The viral titers in the supernatant, as well as the production of viral RNA in epithelial cells from different groups were detected. All data are the mean ± SD, comparisons performed using one-way ANOVA. *$P < 0.05$, **$P < 0.01$. The results are from at least three different experiments.

inhibited by IFN-γ antibody. The above results suggest that the IFN-γ-dependent innate immune response may be their main primary protective mechanism in intestinal viral infection resistance

In conclusion, our findings revealed a unique transepithelial migration of porcine intestinal IELs in providing host-protective immune surveillance. During intestinal PEDV infection, the intraepithelial and transepithelial migration of IELs was promoted by chemokines from virus-infected IECs. Meanwhile, we also identified the antiviral activity of intestinal IELs, which exert potent innate antiviral resistance depending on their IFN-γ production mediated by both viral pre-activation and the signaling of virus-infected IECs. However, the detailed mechanism of the IELs pre-activated by the virus as well as their further activation by virus-infected epithelial cells remain unclear, requiring further investigation (Fig. 9). Our results revealed the distinct function of porcine IELs in the defense against intestinal PEDV infection and shed light on inducing highly effective intestinal mucosal immune protection.

## Methods

**Cells, antibodies, and reagents.** Marc-145 cells and Vero E6 cell lines were preserved in our laboratory. The IPEC-J2 cell lines used in this study were gifted by Professor Wei Zhanyong (Henan Agricultural University). The cells were cultured in Dulbecco's modified Eagle medium (DMEM) with high glucose (Gibco, Gaithersburg, MD, USA), supplemented with 10% fetal bovine serum (FBS; Life Technologies, Grand Island, NY, USA), and incubated in an atmosphere of 5% $CO_2$ at 37 °C. All cell lines were regularly tested for mycoplasma contamination. For intestinal IELs observation and analysis, we used anti-pig PerCP-Cy5.5-CD3ε (BD Biosciences,1:100, 561478), anti-Pig PE-TCR γδ (BD Biosciences, 1:25, 561486), anti-Pig PE-Cy7-CD4 (BD Biosciences, 1:100, 561473), anti-Pig FITC-CD8α (BD Biosciences, 1:100, 551303), and anti-pig CD3 (Abcam, 1:200, ab16669) antibodies. To detect intestinal CCL2 secretion, we used an anti-pig CCL2 mAb antibody (Bioss Antibodies, bs-1955R, 1:100). To detect PEDV protein expression, we used the anti-PEDV N protein mAb (Medgene labs, SD17-103, IFA, 1:200; Western-blot, 1:1000). The other antibodies included anti-mouse tight junction protein protein zonula occludens protein (ZO-1) (Invitrogen, 1:200, Z01-1A12), anti-GAPDH Monoclonal antibody (Solarbio, 1:1000; K200057M) and secondary antibodies used for western blot and IFA, such as goat anti-mouse IgG H&L (HRP) (Abcam, 1:5000; ab205719), goat anti-mouse IgG H&L (Alexa Fluor 488) (Abcam, 1:200, ab150113) and goat anti-rabbit IgG H&L (Alexa Fluor 488) (Abcam, 1:200, ab150077), goat anti-mouse IgG H&L (Alexa Fluor® 594) (Abcam, 1:200, ab150116). For cellular nucleus staining, we used Hoechst 33342 (Life Technologies, 1:2000, H3570). For dead cell staining we used Zombie NIR™ Fixable Viability Kit (BioLegend, San Diego, CA, USA). We used recombinant pig CCL2 (GenScript, Jiangsu, China) and CCL5 protein (R&D, Minneapolis, MN, USA). To inhibit the CCR2 receptor activity of IELs, we used CCR2 antagonist 4 (Teijin compound 1, TC1; MedChemExpress, China). To inhibit perforin/granzyme mediated cytotoxicity of IELs, we used perforin inhibitor concanamycin A (CMA) (Sigma-Aldrich, HY-N1724) or granzyme inhibitor 3,4 Dichloroisocoumarin (DCI) (TargetMol, T8677).

**Viruses and infection.** The PEDV strain, ZJ, was isolated from piglets with severe diarrhea in China in 2012 and was found to cluster with the emerging virulent strain in a previous phylogenetic analysis[41]. The viruses were purified using discontinuous sucrose density gradient centrifugation. WI PEDV was prepared at 56 °C for 0.5 h and tested for complete loss of infectivity by being inoculated into Vero cells at a multiplicity of infection (MOI) = 1 for CPE observations. For PEDV infection, 70% Confluent Marc 145 cells were inoculated with PEDV at a MOI of 0.1 for 1 h at 37 °C. The inoculum and unattached virus were removed, and a fresh growth medium was added. Infected cells were analyzed or co-cultured with IELs/IPEC-J2 and IELs at the required incubation period.

**Animal experiments.** The newborn (0 day), one-month-old, and six-month-old Duroc × Landrace × Yorkshire piglets were obtained from a swineherd at the Jiangsu Academy of Agricultural Science and were raised in highly sanitary conditions in the experimental animal center of Nanjing Agricultural University. The swineherd was seronegative for antibodies against PEDV, porcine reproductive and respiratory syndrome virus (PRRSV), porcine respiratory corona virus (PRCV), transmissible gastroenteritis virus (TGEV), and porcine circovirus type 2. Each experimental pig group was housed in a separate room in a high-security isolation facility.

For the PEDV infection experiment, female piglets (one-month-old) with similar weight were divided into two groups (3 piglets per group) with a completely random design and housed in three separate rooms 24 h prior to the experiment (acclimation period). Piglets of the PEDV challenge group were inoculated with 1 mL PEDV ($10^7$ PFU/ml). The uninfected control piglets were orally administered the same PBS volume. After a challenge, the piglets were observed daily for symptoms of diarrhea. All piglets were euthanized with pentobarbital sodium (100 mg/kg) at 48 h postinfection (hpi), and intestinal samples were collected. For the ligated loop experiments, three 1-month-old female piglets were anesthetized with pentobarbital sodium at a dose rate of 20 mg/kg body weight, and a midline incision was made just anterior to the navel[42]. The terminal jejunal and ileal ligated loops of piglets were injected with three treatments (n = 3): PEDV ($10^6$ PFU/segment) for 1 or 3 h, as well as the same volume of PBS (0.5 mL/segment) and *B. subtilis* (GenBank: MN809319) ($10^7$ CFU/segment)[43], which were used as the negative or positive control, respectively. During the procedure, the piglets were kept warm on a warming pad (37 °C). Three hours postinjection, the intestines were removed, fixed with Bonn's liquid, embedded in paraffin, and cut into 4-µm sections for immunohistochemical and IFA as described below.

**Generation of porcine small intestinal IELs.** IELs were isolated from porcine ileum tissue as previously described[44]. Briefly, the small intestines were opened, freed of Peyer's patches, and washed in PBS. Mucus dissociation was performed by incubating tissues in 30 mL of Hank's balanced salt solution (HBSS) containing 5 mM dithiothreitol (Invitrogen 15508) and 2% heat-inactivated fetal calf serum (FCS; Gibco A38401) for 20 min. The epithelial layer was gently scraped off using a scalpel and incubated for 40 min in RPMI 1640 containing 10% FCS and 1 mM dithiothreitol in a turning wheel. The cells in the epithelial layer were collected by transferring tissue into 30 mL of HBSS containing 5 mM EDTA and 2% FCS. A total of 3 sequential incubations in fresh epithelial removal solution were performed for 25 min each, transferring the tissue to fresh solution for each incubation. Liberated cells from the epithelial removal and wash solutions were retained, pooled, passed through a 100-micron nylon filter, and washed with HBSS containing 2 mM L-glutamine and 2% FCS. Isolated cells were centrifuged in a 20%/40%/80% Percoll density gradient at 700 g for 30 min. The IELs were harvested from the 40% to 80% Percoll interface. After two days, the cells were transferred to fresh plates and cultured with IL-2 (10 U/mL) in RPMI 1640 Supplemented with 10% FBS, 100 U/mL penicillin, 100 µg/mL streptomycin, 2 mM glutamine, 1 mM sodium pyruvate, 2.5 mM HEPES, and non-essential amino acids (Invitrogen). Cells were maintained in 96-well round-bottom plates at concentrations of 500,000 cells/Ml at 37 °C and 10% $CO_2$. The medium was replaced completely every 3–4 days. The viability and quantity of the final epithelial-enriched cell suspensions were determined using the cell counter (JIMBIO CL) with trypan blue dye.

**IELs activation and blocking.** For in vitro activation, $10^5$ IELs per well in flat-bottom 96-well microtiter plates were incubated with whole inactivated PEDV (MOI = 0.1) or recombinant expressed spike protein of PEDV (50 ng) and centrifuged at 1200 × g for 15 min. Next, the cell pellet was re-suspended and maintained at 37 °C for 1 h. For in vivo IELs activation, six piglets (one-month-old) were divided into two groups, inoculated with either 1 mL PBS or whole inactivated PEDV ($10^7$ PFU/mL). Four-hour postoral inoculation, the ileum tissues were collected for IELs isolation. In IELs receptor blocking experiments, IELs were pre-incubated with CCR2 antagonist TC1 for 12 h before being co-cultured with PEDV-infected Marc 145 cells. To neutralize the IFN-γ secreted by IELs, antibodies against porcine IFN-γ (R4-6A2; R&D) were supplemented when the IELs were co-cultured with PEDV-infected Marc 145 cells.

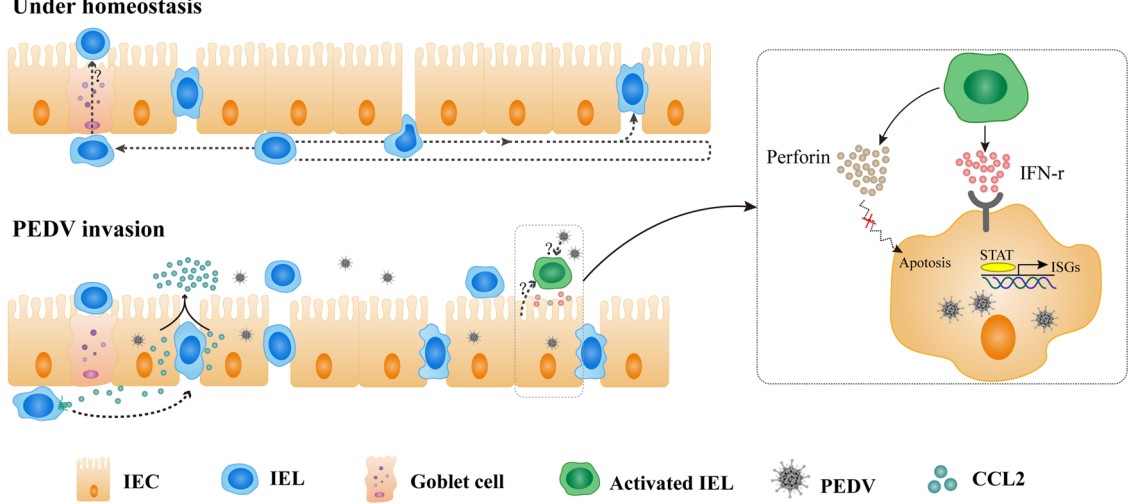

**Fig. 9 Schematic of the proposed mechanism of intestinal intraepithelial lymphocytes (IELs) functions in response to intestinal porcine epidemic diarrhea virus (PEDV) infection.** Under homeostatic conditions, porcine intestinal IELs not only exhibit a flossing behavior in which the cells move into the LIS between two adjacent enterocytes, but also move to the free end of the intestinal epithelia to perform their daily immune surveillance behavior. During PEDV infection, IELs were recruited by the polarity distribution of CCL2 secreted from virus-infected intestinal epithelial cells (IECs). The IELs' enhanced intraepithelial and transepithelial migration may benefit their interaction upon encountering viral antigens. Intestinal IELs could exert potent innate antiviral resistance by viral pre-activation and the signaling of virus-infected IECs. However, the integral mechanisms involved in this process are not well elucidated. Although the secretion of perforin from IELs was significantly increased, the antiviral function of activated IELs was mainly mediated by IFN-γ secretion that induced robust antiviral response in virus-infected cells.

**Establishment of the IELs/IPEC-J2, IELs/IPEC-J2/Mac 145, and IELs/Mac 145 co-culture system**. After turning upside down the culture inserts of Transwell (0.4 μm pore size; 6.5 mm membrane diameter; Corning, Corning, NY, USA) into a 12-well plate, the collected IPEC-J2 cells were seeded at a concentration of $2 \times 10^5$ cells/cm onto the backside of the membrane of Transwell inserts for 8 h to allow the cells attach to the filter. Subsequently, the filters were then turned upside-down again into a 24-well plate and maintained for 5 days until steady-state TEER of 500 $\Omega \times cm^{-2}$ was achieved. Thereafter, the isolated IELs were seeded on the baso-lateral side of IPEC-J2 (upper chamber) and co-cultured for the indicated time. In the process of co-culturing, the change of TEER and expression of tight junction protein ZO-1 was determined to evaluate the influence of IECs on the epithelial barrier integrity of IPEC-J2.

Marc 145 cells were seeded in 24-well plates and grown to monolayers before establishing IELs/IEPC-J2/Mac 145 or IELs/Mac 145 cells co-culture systems. In the former co-culture system, the Transwell assemblies including IELs/IPEC-J2 were cultured over Marc 145 cell monolayers without direct contact, whereas in the latter co-culture system, the IELs were directly cultured with Marc 145 cells. IELs, Mac 145 cells and the medium from each co-culture were collected separately at the indicated times.

**Plaque assay and western blot analysis**. Confluent monolayers of Vero E6 cells grown in 6-well tissue culture plates were infected with 500 μL of serial tenfold dilutions of the supernatant samples. After being incubated for 1 h at 37 °C, cells were overlaid with 0.7% Sea-Plaque agarose in DMEM containing 2% FBS and incubated at 37 °C. Three days post-infection, the plaques were visualized via crystal violet staining.

Cell samples were assessed using western blotting with specific antibodies. At indicated times post-infection, the collected cell samples were lysed in RIPA buffer containing a protease inhibitor cocktail (Thermo Scientific). The concentration of the lysates was determined using a BCA protein quantification kit (Thermo Scientific). Equal amounts of protein were separated via SDS-PAGE and electrophoretically transferred onto a PVDF membrane (Millipore, Shanghai, China). The membranes were blocked with 5% skim milk for 2 h and incubated with specific primary antibodies, followed by incubation with appropriate horseradish peroxidase-conjugated secondary antibodies. The intensity of the bands in terms of density was measured and normalized against glyceraldehyde 3-phosphate dehydrogenase (GAPDH) expression. Uncropped western blot images of data can be found in Supplementary Fig. 9.

**Flow cytometric analysis and separation**. The isolated and purified IELs were stained with specific phenotype antibodies to analyze the IELs populations. The collected IELs were counted and stained with Fixable Viability Dye (Zombie NIR™ Fixable Viability Kit) according to the manufacturer's recommendations. Next, sequential incubations with conjugated primary antibodies included CD3ε- PerCP-Cy5.5, CD4-PE-CY7, CD2-APC, γδT-PE and CD8-FITC. After antibody staining, the cells were resuspended, passed through a 35-micron nylon filter to remove

aggregates, and data were acquired using BD FACS Verse. The instrument was set up according to the manufacturer's recommendations using bead capture reagents to set compensation controls.

**Quantitative RT-PCR**. Total RNA from IELs and Mac 145 cells was extracted using a TRIzol reagent (Invitrogen) according to the manufacturer's instructions. cDNA was generated by reverse transcription using HiScript™ Q RT SuperMix for qPCR (Vazyme, China) according to the manufacturer's instructions. The target gene transcription was determined using quantitative RT-PCR (RT-qPCR) with the SYBR Green qPCR Kit (Takara, Beijing, China) and analyzed using the double standard curve method. All primers used for RT-qPCR are presented in Supplementary Table 1. Cellular GAPDH was quantified as the internal control. The relative levels of cytokine RNA were calculated using the $2^{-\Delta\Delta CT}$ method.

**Immunohistochemistry**. Small intestinal tissue was immersed in Bouin's solution for 36 h before being embedded in paraffin wax. Paraffin-embedded sections were transformed into serial sections with a thickness of 5–8 mm using a microtome (Leica RM2235). Paraffin sections were dewaxed in xylene and rehydrated in decreasing concentrations of ethanol. For IHC analysis, antigen retrieval was performed for 30 min with citrate buffer at pH 6.0 in a Decloaking Chamber at 95 °C. Slides were blocked with 5% bovine serum albumin (BSA) and then incubated with anti-pig CD3 IgG (Abcam, 1:200, ab16669) overnight at 4 °C in a humidified chamber. The SABC-POD Kit was used for the amplification and visualization of the signal. The sections in which primary antibodies were omitted were used as negative controls. The numbers of IELs at 10 different fields in the ileum villus in each piglet were counted for the statistical analysis.

**Immunofluorescence and confocal microscopy**. To observe the CCL2 secretion in the ileum tissue, tissue sections were rinsed and subjected to antigen demasking as described above. After rinsing in PBS, sections were treated with 5% BSA for 20 min and incubated with mouse anti-pig CCL2 overnight at 4 °C, followed by incubation with fluorescent secondary antibodies for 1 h at 25 °C. The negative control slides were treated identically, except for the removal of the primary antibodies. For tight junction observation, IPEC-J2 cells were labeled with ZO-1 mAbs followed by Alexa Fluor 594- conjugated goat anti-mouse IgG. Cell nuclei were stained by incubation with diamidino-2-phenylindole (DAPI) for 5 min, and observed under a confocal laser microscope (LSM-710; Zeiss).

For the cell migration assay, the IELs were labeled with CFSE (green) before being co-cultured with IPEC-J2. Next, the IELs/IPEC-J2 co-culture system was subject to different treatments according to the experimental requirements and fixed in 4% paraformaldehyde. Fixed filters were permeabilized in 0.2% Triton X-100 in PBS for 5 min. After being blocked with 5% bovine serum albumin in PBS for 1 h, the filters were incubated with the primary antibodies overnight at 4 °C, followed by fluorescent secondary antibodies at room temperature for 1 h. The filters were visualized via CLSM (LSM 710, Zeiss). Serial sections were collected

using Z-project with a 0.5 mm increment on the z-axis. Cross-section x-z images were rendered using Zeiss ZEN 2012 software. Three-dimensional (3D) rendering of representative fields was obtained with Imaris 7.2 software.

**Cytokine assays**. The production of cytokines (CCL2, CCL5, IFN-γ, IFN-α, and IFN-β) was measured using an ELISA kit (eBioscience), according to the manufacturer's instructions.

**Statistics and reproducibility**. Results are presented as means ± SD and analyzed with SPSS 17.0. One-way ANOVA was employed to determine statistical differences among multiple groups, and *t*-test was employed to determine differences between the two groups. $*P < 0.05$, $**P < 0.01$. Data were combined from at least three independent experiments unless otherwise stated. The sample size is indicated for each experiment in the corresponding figure legend.

**Ethics statement**. All animal procedures and experiments were performed according to protocols approved by the Institutional Animal Care and Use Committee of Nanjing Agricultural University (Nanjing, China) and followed the National Institutes of Health guidelines.

**Reporting summary**. Further information on research design is available in the Nature Research Reporting Summary linked to this article.

## Data availability

Data supporting the findings of this work are available within the paper and its Supplementary files or are available from the corresponding author upon reasonable request. Raw data underlying plots in the figures are available in Supplementary Data 1. Unprocessed blot images with size markers are provided in Supplementary Information.

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

## Acknowledgements

This work was financially supported by the National Key Research and Development Program of China (Grant No. 2021YFD1801105), the National Natural Science Foundation of China (31930109 and 32002261), the Natural Science Foundation of Jiangsu Province (No. BK20200536), Fundamental Research Funds for the Central Universities (KJQN202133), China Postdoctoral Science Foundation funded project (BX20190153) and the Priority Academic Program Development of Jiangsu Higher Education Institutions (PAPD).

## Author contributions

Conception of the work: Q.Y. and Y.L; Cellular and animal experiment: Y.M., X.P., Y.J., Y.L., X.W., P.Z., P.L., C.L.; Analysis and interpretation of data: Y.L., Y.M., Y.J.; Preparation of the manuscript: Q.Y. and Y.L. All authors read and approved the final manuscript.

## Competing interests

The authors declare no competing interests.
