## [Peer Review File · Communications Biology]

Reviewers' comments:

Reviewer #1 (Remarks to the Author):

The authors present a manuscript documenting the migratory characteristics and anti-viral activity of intraepithelial lymphocytes (IEL) in pigs. IEL constitute a large proportion of the body's T cells but are comparatively understudied, despite displaying important anti-viral and anti-cancer potential. Moreover, IEL exhibit unconventional characteristics, such as innate-like reactivity and surveillance behaviour. While a growing body of work has documented the migratory behaviour and anti-viral activity of IEL in mice, how this may translate to humans is less clear. However, IEL have been implicated in coeliac disease and IBD, amongst other pathologies, suggesting that more information is needed regarding how these cells function in beneficial and pathological scenarios. Many IEL are $\gamma\delta$ T cells, and pigs and ruminants show distinct $\gamma\delta$ T cell populations from mice, warranting a closer look at IEL activity in these animals. This may shed some light into the basic questions about IEL that we still lack clear answers to, such as antigen specificity and molecular signals regulating their behaviour.

The authors begin by tracking localisation of IEL in intestinal samples from pigs as they mature. They record a previously unappreciated phenomenon, where apparently viable IEL could be found on the apical face of the epithelium i.e. in the intestinal lumen. They also see a steady increase in IEL numbers as pigs develop and are weaned, suggesting an influence from the microbiome. Moreover, following infection with porcine epidemic diarrhea virus (PEDV), an industrially-relevant pathogen, they see a dramatic and rapid increase in the number of T cells in the intraepithelial and luminal compartments. A ligated jejunal loop model is employed here to permit highly localised and acute analysis of this migratory response, which also occurs following bacterial inoculation. To further elucidate this transepithelial migration, the authors develop an in vitro system whereby IEL migration through an epithelial monolayer is observed when a spatially separated epithelial cell line is infected with PEDV. They then show that CCL2 but not CCL5 may induce transepithelial migration of IEL in this system, and that inhibition of CCL2 largely blocks their migration in response to PEDV infection.

The second half of the manuscript instead investigates the effector function of porcine IEL in in vitro PEDV infection. Contrary to mice, where anti-CD3 stimulation promoted anti-viral activity of IEL against norovirus, the authors saw no impact of unstimulated or anti-CD3 pre-treated porcine IEL upon PEDV replication in an in vitro co-culture assay. However, pre-treatment of IEL with whole inactivated (WI) PEDV induced a dramatic anti-viral activity in IEL upon subsequent co-culture with a live PEDV-infected cell line. This anti-viral activity was contact-independent as shown by transwell assay. A similar effect was seen with IEL pre-activated by in vivo WI PEDV infection. Finally, the authors investigate the anti-viral activity in more detail. WI PEDV-pre-activated IEL reduced overall PEDV infection of the cell line, but increased apoptosis of infected cells, alongside a large induction of perforin in the IEL. More mechanistically, they find a clear induction of IFN- γ in pre-activated IEL upon co-culture with infected cells, which correlated with the induction of interferon-stimulated genes in the infected cells and led to control of infection as demonstrated by inhibition of IFN- γ .

Overall, these findings are relevant to the IEL field as they are the first robust characterisation of porcine IEL behaviour at steady state and following viral infection. Given the similarities but also key differences between IEL in mouse and humans, better study of IEL in more closely related species will help address important questions relevant to human health and disease. The authors generally use the best techniques available, considering the obvious technical limitations of live imaging in pigs, and the necessity for in vitro assays in certain contexts. The observation of luminal T cells is intriguing, although probably needs to be better characterised in the future to ensure it is not an artefact of intestinal harvesting, which is prone to autolysis. Moreover, the strict requirement for acute pre-exposure to inactivated PEDV (either in vitro or in vivo) to activate IEL, which then only perform anti-viral activity in the presence of live PEDV-infected cells, suggests some interesting biology which may have implications for IEL responses in many systems. The authors should discuss these implications, namely that IEL appear to require microbe pre-exposure (over an extremely short timeframe) in order to induce a capacity to respond to what appears to be host-derived soluble signals following infection with a live virus. Moreover, there should be

discussion of how the PEDV may activate the IEL, especially if anti-CD3 TCR stimulation did not have an equivalent effect.

Major concerns

1. My predominant concern with the manuscript is in the authors' conclusions, namely that the observed migratory behaviour of IEL is contributing to their anti-viral function. This is an extremely tenuous link, purely based upon the observation that a small percentage of IEL enter the lumen following PEDV infection, where it is possible that they directly sense PEDV, which may then link to the requirement for PEDV pre-exposure for the anti-viral assays. However, the authors repeat this often, and even in the title of the manuscript. While it is likely that migratory behaviour of IEL contributes to anti-viral surveillance (as reported by Hoytema van Konijnenburg et al Cell 2017, amongst others), this is not supported by the authors' experiments. The authors in effect have a manuscript describing IEL migration in viral infection, and separately, anti-viral function by IEL. They need to tone down these conclusions and amend the title.

2. My other concern regards the interpretation of the WI PEDV pre-activation. Apparently, this occurs over the course of 1hr culture of IEL with WI PEDV, prior to the IEL being transferred into contact with a live PEDV-infected cell line. However, IEL that were not first cultured with WI PEDV displayed no anti-viral effect upon co-culture with infected cells, despite the fact that PEDV was obviously present in these co-cultures (given plaque assays using culture S/N e.g. Fig 5B). I wonder if the authors can explain how 1hr contact with WI PEDV, but not 24hr contact with PEDV in the S/N of infected cells, resulted in IEL activation?

3. Throughout the paper, the IEL are treated as bulk, except in Figure 7A where $\gamma\delta$ T cells are suddenly stained for no clear reason. It would be helpful if the authors could investigate, at least, whether $\alpha\beta$ vs. $\gamma\delta$ IEL show similar anti-viral behaviour, and requirement for WI PEDV pre-activation. It seems that amongst porcine IEL, $\gamma\delta$ T cells express the most NKG2D, which might be relevant to the perforin/cytolysis data in Figure 7 (Altmeyer et al 2017 Vet Immunol Immunopathol). In addition, it might add some useful data to ask whether the perforin/cytolysis results occur in the transwell system, where NKG2D-ligand interactions between IEL and infected cells would be prevented.

Minor concerns

4. The CCL2 experiments in Figure 4 are independently valid. However, the authors should reconcile the fact that WI virus does not induce CCL2 in their cell line with the observations of IEL migration changes following in vivo WI virus administration in Figure 5. Perhaps explaining this, Fig 4B suggests that WI PEDV introduction into the intestine causes CCL2 upregulation, but this is not confirmed by the figure legend. Was this in fact live or WI virus?

5. Perhaps the IEL do not migrate to CCL5 because they simply do not express CCR5? This could be easily investigated by qPCR if porcine antibodies are not available.

6. Figures 1 and 2 show summary data without error bars or statistics on the plots, despite p-values being defined in the figure legends and in the text. These should be shown on the graphs (Fig 1F, Fig 2B, Fig 2E). Moreover, the number of animals used in these experiments should be included in all figure legends.

7. The quantification of T cell localisation to basal, intraepithelial and luminal areas in Figures 1 and 2 is helpful as absolute number, but frequency of total T cells in each area (%) should also be presented. This will provide evidence for the authors' claims that the frequency of luminal or intraepithelial-located T cells changes upon infection/ageing. Currently, the absolute counts merely suggest an overall influx of T cells, without a specific relocation. Additionally, basal T cells should be quantified in Figure 2.

8. The multiple comparisons tests used, for example, in Figure 4F and Figure 5D, F G and H) are a little unclear. Are they truly comparing all groups, or are some groups combined before comparison (i.e. the brackets linking two columns). If they simply summarise equivalent p-values for the comparison between one group and two other groups, the individual comparisons should be displayed.

9. The flow cytometric staining of $\gamma\delta$ T cells in Figure 7A is not convincing. The gating looks arbitrary. $\gamma\delta$ TCR vs $\alpha\beta$ TCR staining should be shown on the same plot in order to convincingly discriminate between the two populations.

10. Figure legends need to have better information regarding timepoints, particularly in the case of in vivo PEDV infection i.e. how long after infection are animals in Figure 1 examined?

Reviewer #2 (Remarks to the Author):

The current study provides evidence of transepithelial migration of IELs in porcine and how this novel migratory behavior could protect against oral infection by PEDV. Finally, the study concludes that protection against PEDV is mediated by IFN gamma and the cytotoxic nature of activated IELs. Though these findings have some novelty, they are somewhat preliminary and are not backed by solid evidence.

Specific comments -

1. Though authors found IELs on the luminal side, there is no direct evidence that these IELs leaked from the damage site of the epithelium layer.

2. Current study uses the IPEC-J2-IEL co-culture system as a model to decipher the mechanisms invitro. However, in all of the experiments, IELs are cultured on the apical side of the IPEC-J2 cells. This is opposite to what happens invivo, where IELs are present on the basal side of the epithelium.

3. Further, it was shown that upon PEDV infection, CCL2 was mainly present on the apical surface, which acts as a chemokine for the transepithelial migration of IELs. Nevertheless, in the IPEC-J2-IEL system, IELs are already on the apical side and then moving towards the basal side. How do the authors corroborate these two observations?

4. Next, it was found that IELs are can be directly activated by WI - PEDV. How are T cells detecting the virus? Do porcine IELS/T cells express TLRs or other pathogen recognition receptors (PRRs) detecting the virus? Similarly, how do mLN T cells are getting activated by invitro stimulation by PEDV?

5. One possibility by which IELs could be activated would be contamination of sorted IELs by epithelial cells, which express various PRRs. To ensure this is not the case, can authors provide pre and post sort profiles of the IELs?

6. In panel 5F, why are there no plaques seen in the mock of direct co-culture?

7. In Fig 7, it was concluded that granzymes and perforin mediated antiviral activity. However, this is just a correlation as Gzms and perforins were only measured by ELISA, and no direct evidence is provided that these molecules are mediating the death of epithelial cells. Moreover, apoptosis of epithelial cells can also be caused by IFN gamma.

8. In conclusion, the authors found a transepithelial behavior of porcine IELs, but its implications are not very clear from the current data presented in the manuscript. Therefore, the paper's title seems to be an overstatement in concluding that transepithelial migration leads to the antiviral state.

Manuscript COMMSBIO-21-1856 R1

Title: The migration behavior and innate immune function of intraepithelial lymphocytes response to intestinal virus infection

Point to point response

Reviewers' comments:

Reviewer 1: The authors present a manuscript documenting the migratory characteristics and anti-viral activity of intraepithelial lymphocytes (IEL) in pigs. IEL constitute a large proportion of the body's T cells but are comparatively understudied, despite displaying important anti-viral and anti-cancer potential. Moreover, IEL exhibit unconventional characteristics, such as innate-like reactivity and surveillance behaviour. While a growing body of work has documented the migratory behaviour and anti-viral activity of IEL in mice, how this may translate to humans is less clear. However, IEL have been implicated in coeliac disease and IBD, amongst other pathologies, suggesting that more information is needed regarding how these cells function in beneficial and pathological scenarios. Many IEL are $\gamma\delta$ T cells, and pigs and ruminants show distinct $\gamma\delta$ T cell populations from mice, warranting a closer look at IEL activity in these animals. This may shed some light into the basic questions about IEL that we still lack clear answers to, such as antigen specificity and molecular signals regulating their behaviour. The authors begin by tracking localization of IEL in intestinal samples from pigs as they mature. They record a previously unappreciated phenomenon, where apparently viable IEL could be found on the apical face of the epithelium i.e. in the intestinal lumen. They also see a steady increase in IEL numbers as pigs develop and are weaned, suggesting an influence from the microbiome. Moreover, following infection with porcine epidemic diarrhea virus (PEDV), an industrially-relevant pathogen, they see a dramatic and rapid increase in the number of T cells in the intraepithelial and luminal compartments. A ligated jejunal loop model is employed here to permit highly localised and acute analysis of this migratory response, which also occurs following bacterial inoculation. To further elucidate this transepithelial migration, the authors develop an in vitro system whereby IEL migration through an epithelial monolayer is observed when a

spatially separated epithelial cell line is infected with PEDV. They then show that CCL2 but not CCL5 may induce transepithelial migration of IEL in this system, and that inhibition of CCL2 largely blocks their migration in response to PEDV infection. The second half of the manuscript instead investigates the effector function of porcine IEL in in vitro PEDV infection. Contrary to mice, where anti-CD3 stimulation promoted anti-viral activity of IEL against norovirus, the authors saw no impact of unstimulated or anti-CD3 pre-treated porcine IEL upon PEDV replication in an in vitro co-culture assay. However, pre-treatment of IEL with whole inactivated (WI) PEDV induced a dramatic anti-viral activity in IEL upon subsequent co-culture with a live PEDV-infected cell line. This anti-viral activity was contact-independent as shown by transwell assay. A similar effect was seen with IEL pre-activated by in vivo WI PEDV infection. Finally, the authors investigate the anti-viral activity in more detail. WI PEDV-pre-activated IEL reduced overall PEDV infection of the cell line, but increased apoptosis of infected cells, alongside a large induction of perforin in the IEL. More mechanistically, they find a clear induction of IFN- γ in pre-activated IEL upon co-culture with infected cells, which correlated with the induction of interferon stimulated genes in the infected cells and led to control of infection as demonstrated by inhibition of IFN- γ . Overall, these findings are relevant to the IEL field as they are the first robust characterisation of porcine IEL behaviour at steady state and following viral infection. Given the similarities but also key differences between IEL in mouse and humans, better study of IEL in more closely related species will help address important questions relevant to human health and disease. The authors generally use the best techniques available, considering the obvious technical limitations of live imaging in pigs, and the necessity for in vitro assays in certain contexts. The observation of luminal T cells is intriguing, although probably needs to be better characterised in the future to ensure it is not an artefact of intestinal harvesting, which is prone to autolysis. Moreover, the strict requirement for acute pre-exposure to inactivated PEDV (either in vitro or in vivo) to activate IEL, which then only perform anti-viral activity in the presence of live PEDV-infected cells, suggests some interesting biology which may have implications for IEL responses in many systems. The authors should discuss these implications, namely that IEL appear to require

microbe pre-exposure (over an extremely short timeframe) in order to induce a capacity to respond to what appears to be host-derived soluble signals following infection with a live virus. Moreover, there should be discussion of how the PEDV may activate the IEL, especially if anti-CD3 TCR stimulation did not have an equivalent effect.

A: Thank you very much for your insightful comments and thorough analysis of our manuscript. We acknowledge and welcome your suggestions, which will be incorporated to improve the quality of our manuscript. We will respond to your specific questions one by one, perform the relevant experiments and correct a series of errors found in the manuscript. All the changes made by us are labeled with red font in the revised manuscript.

Q1. My predominant concern with the manuscript is in the authors' conclusions, namely that the observed migratory behaviour of IEL is contributing to their anti-viral function. This is an extremely tenuous link, purely based upon the observation that a small percentage of IEL enter the lumen following PEDV infection, where it is possible that they directly sense PEDV, which may then link to the requirement for PEDV pre-exposure for the anti-viral assays. However, the authors repeat this often, and even in the title of the manuscript. While it is likely that migratory behaviour of IEL contributes to anti-viral surveillance (as reported by Hoytema van Konijnenburg et al Cell 2017, amongst others), this is not supported by the authors' experiments. The authors in effect have a manuscript describing IEL migration in viral infection, and separately, anti-viral function by IEL. They need to tone down these conclusions and amend the title.

A1: Thank you for your careful review and thoughtful comments. Although the transepithelial migration of IELs may favor the immediate sense of PEDV and pre-activation, the effects of such migratory behavior on their antiviral activity remain speculative and need to be verified in further studies. We apologize for our inaccurate conclusions. The relevant sections in the conclusions and the title have been edited in the revised manuscript.

Q2. My other concern regards the interpretation of the WI PEDV pre-activation. Apparently, this occurs over the course of 1hr culture of IEL with WI PEDV, prior to the IEL being transferred into contact with a live PEDV-infected cell line. However, IEL that were not first cultured with WI PEDV displayed no anti-viral effect upon co-culture with infected cells, despite the fact that PEDV was obviously present in these co-cultures (given plaque assays using culture S/N e.g. Fig 5B). I wonder if the authors can explain how 1hr contact with WI PEDV, but not 24hr contact with PEDV in the S/N of infected cells, resulted in IEL activation?

A2: Your careful review is much appreciated. We apologize for the missing description for IELs pre-activation. In fact, incubation with inactivated PEDV alone cannot induce IELs' antiviral potential (Fig. R1). It was only when we added a centrifugation step (IELs and WI PEDV were centrifuged at 1200×g for 15 min at 25 °C) before incubation that the IELs became effectively pre-activated. With regards to why the centrifugal process is critical for the pre-activation of IELs by viral antigens, previous studies have shown that cultured lymphocytes in suspension have a low chance of contacting with virions during virus inoculation, while centrifugation could dramatically enhance the binding of the viral particles to lymphocytes (~40 fold)^{1,2,3}. Based on this observation, we speculated that centrifugation may facilitate the pre-activation of IELs in the incubation phase by increasing the binding of WI PEDV to IELs. However, further studies are needed to confirm this speculation further. Thank you again for your careful review and thought-provoking questions.

Fig.R1. Simple virus incubation does not induce the antiviral activity of IELs.

Mock treated and WI PEDV treated IELs (without centrifugation steps) were co-cultured with PEDV-infected epithelial cells for 24 h. **a** The expression of PEDV-N

protein in epithelial cells was detected using western blot. **b** The intracellular viral RNA levels were quantitated via RT-qPCR. All data are the mean \pm SD, and comparisons were performed using one-way ANOVA. * $P < 0.05$, ** $P < 0.01$. The results are from at least three different experiments.

Q3. Throughout the paper, the IEL are treated as bulk, except in Figure 7A where $\gamma\delta$ T cells are suddenly stained for no clear reason. It would be helpful if the authors could investigate, at least, whether $\alpha\beta$ vs. $\gamma\delta$ IEL show similar anti-viral behaviour, and requirement for WI PEDV pre-activation. It seems that amongst porcine IEL, $\gamma\delta$ T cells express the most NKG2D, which might be relevant to the perforin/cytolysis data in Figure 7 (Altmeyer et al 2017 Vet Immunol Immunopathol). In addition, it might add some useful data to ask whether the perforin/cytolysis results occur in the transwell system, where NKG2D-ligand interactions between IEL and infected cells would be prevented.

A3: Much appreciated for your careful review and constructive suggestion. In Fig.7A, the purified IELs were stained with specific phenotype antibodies to analyze the IELs populations. FACS analysis showed that $CD8\alpha^+$ T cells accounted for the main subgroup of porcine intestinal IELs, which alerted us of the need to explore the cytolytic activity of IELs. I am sorry for this abrupt result, a detailed description about the purpose of this study has been added in the revised manuscript.

At present, we are unable to separate the $\alpha\beta^+$ or $\gamma\delta^+$ IEL from porcine intestinal IELs for the following reasons. 1) There are currently no commercial $\alpha\beta^+$ IEL antibodies available for flow cytometry staining; 2) Although previous study have considered $\gamma\delta^-$ T cells as $\alpha\beta^+$ IEL, the $\gamma\delta^+$ IELs still cannot be effectively sorted out since the grouping of $\gamma\delta$ T positive cells was not obvious after anti-Pig $\gamma\delta$ T lymphocytes antibody staining. (Fig. R2). Based on your kindly suggestions. The LDH release and perforin secretion in the two co-culture models (contact or noncontact) were further determined. As shown in Fig.R3, the pre-activated IELs from the two culture models both exhibited highly perforin secretion and similar cytotoxic responses. Regarding the mechanism underlying this observed phenomena,

we can speculate that although the NKG2D-ligand interactions have been proven to be relevant for activating the cytolytic responses of $\gamma\delta$ T cells, many other activating receptors (without the necessity of cell-to-cell contact) were also involved in triggering the cytotoxic activities of IELs^{5, 6}. For example, gut-resident IELs also expressed high levels of cytotoxicity in activating receptor NKp46, which bound the ligands with various viral and bacterial proteins. In addition, the expression of OX40 in CD8⁺T IEL was also associated with T-IEL-mediated cytotoxicity. As the ligand of OX40, OX40L was mostly expressed on the T and B lymphocytes, rather than on the intestinal epithelial cells⁹.

Fig.R2. $\gamma\delta^+$ IELs were sorted by flow cytometry. After percoll purification, the $\gamma\delta^+$ IELs were sorted by FACS (BD FACSAria Flow Cytometer). The purity of the input and sorted cells was detected by flow cytometry.

Fig.R3. The cytotoxic activity of IELs in the two different co-culture systems. The whole inactivated PEDV pre-activated or unstimulated IELs that were co-cultured with epithelial cells using two methods (contact and noncontact co-culture). a Levels of perforin release in the culture supernatants of different groups were measured via

ELISA. b The lactate dehydrogenase (LDH) release was used to evaluate the cytotoxicity of IELs in different groups.

Q4. The CCL2 experiments in Figure 4 are independently valid. However, the authors should reconcile the fact that WI virus does not induce CCL2 in their cell line with the observations of IEL migration changes following *in vivo* WI virus administration in Figure 5. Perhaps explaining this, Fig 4B suggests that WI PEDV introduction into the intestine causes CCL2 upregulation, but this is not confirmed by the figure legend. Was this in fact live or WI virus?

A4: Thank you for your kind suggestions. I apologize for our mistake. Actually, the intestinal section used in Fig 4B was obtained from the piglets injected with live PEDV (1 h), which was described in Fig 2C. This mistake has been corrected in the revised Fig.4.

WI PEDV administration also promoted the entry of IELs into the intestinal epithelial layer, which was inconsistent with the results from the *in vitro* co-culture system. As for the inconsistent results between the *in vitro* and *in vivo* experiments, we speculate that because the cell composition of the co-culture system is simple, when the WI virus cannot induce the intestinal epithelial cells (IECs) to secrete CCL2, then the migration behavior of the IELs will not be triggered. However, in the intestinal mucosa, although the inactivated virus could not regulate the behavior of IELs by affecting epithelial cells, it may promote the migration of IELs by activating some antigen presenting cells. For instance, both the specialized intestinal epithelial cells or microfold cells (M cells), and the migratory CD103⁺ dendritic cells (DCs) (could form transepithelial dendrites (TEDs) sample luminal antigen) all have the chance to come in contact with the antigen and be activated^{10, 11}, which would in turn initiate a cascade of IELs responses^{6, 12}. Additionally, the cell lines used *in vitro* could not incorrectly reflect the *in vivo* situation, which may also responsible for this phenomenon. However, such an explanation is only speculative and requires clarification through further studies.

Q5 Perhaps the IEL do not migrate to CCL5 because they simply do not express CCR5? This could be easily investigated by qPCR if porcine antibodies are not available.

A5: Much appreciated for your kindly comments. Based on your suggestion, we further investigated the gene expression of CCR5 in porcine intestinal IELs. The primers used for RT-qPCR are presented in Table R1. Unexpectedly, the CCR5 gene was also expressed in the porcine intestinal IELs, although its transcription level was lower than the level of CCL2 in IELs (Fig. R4). However, there remains some uncertainty as to whether the CCR5 protein could really be expressed and distributed in the surface of porcine IELs due to a lack of appropriate commercial antibodies. This issue still requires further exploration in future studies.

Fig.R4. The transcriptional level of CCR2 and CCR5 in intestinal IELs.

a, b The expression of CCR2 and CCR5 mRNAs in intestinal IELs were analyzed by RT-PCR (a) or quantitative real-time PCR (b).

Q6. Figures 1 and 2 show summary data without error bars or statistics on the plots, despite *p*-values being defined in the figure legends and in the text. These should be shown on the graphs (Fig 1F, Fig 2B, Fig 2E). Moreover, the number of animals used in these experiments should be included in all figure legends.

A6: Thank you for pointing this out. We apologize for the missing information. We have added the error bars accordingly in the revised figures (including Fig 1F, Fig 2B, Fig 2E) and different letters to indicate significant differences. Moreover, all figure legends have also described the number of animals used. Thank you again for your careful review.

Q7. The quantification of T cell localisation to basal, intraepithelial and luminal areas in Figures 1 and 2 is helpful as absolute number, but frequency of total T cells in each area (%) should also be presented. This will provide evidence for the authors' claims that the frequency of luminal or intraepithelial-located T cells changes upon infection/ageing. Currently, the absolute counts merely suggest an overall influx of T cells, without a specific relocalisation. Additionally, basal T cells should be quantified in Figure 2.

A7. Thank you for your kindly reminder. We have added the frequency of total T cells (%) in each area of Fig. 1F, Fig. 2B, and 2E. Moreover, the number of basal T cells in Fig. 2B and 2E was also be quantified.

Q8. The multiple comparisons tests used, for example, in Figure 4F and Figure 5D, F G and H) are a little unclear. Are they truly comparing all groups, or are some groups combined before comparison (i.e. the brackets linking two columns)? If they simply summarise equivalent p-values for the comparison between one group and two other groups, the individual comparisons should be displayed.

A8. Thank you for pointing this out. We apologize for this mistake. To clarify this error, we have corrected the display form of multiple comparisons and individual comparisons in the revised figures (including Fig.3d, Fig.3e, Fig.4c, Fig.4f, Fig.5c, Fig.5d, Fig.5f, Fig.5g, Fig.5h, Fig.6e, Fig.6f, Fig.7e Fig.7f, Fig.8d, Fig.8e, Fig.8f, Fig.8g, Fig.8h, Fig.8i and Fig.S4). Thank you again for your careful review.

Q9. The flow cytometric staining of $\gamma\delta$ T cells in Figure 7A is not convincing. The gating looks arbitrary. $\gamma\delta$ TCR vs $\alpha\beta$ TCR staining should be shown on the same plot in order to convincingly discriminate between the two populations.

A9: We welcome and acknowledge your suggestion. The PE-labeled rat anti-Pig $\gamma\delta$ T lymphocytes antibody (Cat. No. 561486; BD Biosciences, USA) is the only commercially available antibody for pig $\gamma\delta$ T cells. When this antibody is used for staining, the population of $\gamma\delta$ T positive cells cannot be clearly detected, although the positive cells were markedly shifted to the right. We have provided the FMO-isotype

control to show our gating strategy in the revised Fig.3A (Fig. R5). it was not possible to determine the $\alpha\beta$ TCR positive cells in porcine IELs in our present study.

Fig.R5. Flow cytometry analysis revealed population and subpopulations of isolated porcine intestinal IELs. Fluorescence-minus-one (FMO) isotype-controls determined the boundary between $\gamma\delta$ TCR positive and negative cells.

Q10. Figure legends need to have better information regarding timepoints, particularly in the case of in vivo PEDV infection i.e. how long after infection are animals in Figure 1 examined?

A10: Thank you for your careful review. We apologize for the missing details. We have added the timepoints information in the figure legend of the revised manuscript.

Reviewer 2: The current study provides evidence of transepithelial migration of IELs in porcine and how this novel migratory behavior could protect against oral infection by PEDV. Finally, the study concludes that protection against PEDV is mediated by IFN gamma and the cytotoxic nature of activated IELs. Though these findings have some novelty, they are somewhat preliminary and are not backed by solid evidence. Thank you for your careful review and comments. We acknowledge and welcome your suggestions, which will be incorporated to improve the quality of our manuscript. We apologize for the imperfect study design and over-interpreting research results. At present, we have responded to your specific questions individually. As per your suggestions, we have further performed the relevant supplementary experiments and rewritten the relevant content in the "Discussion" and "Conclusion" sections. All the changes we made are labeled with red font in the revised manuscript.

Q1. Though authors found IELs on the luminal side, there is no direct evidence that these IELs leaked from the damage site of the epithelium layer.

A1. Thank you for your careful review. I apologize for the misunderstanding caused by our inadequate interpretation. Indeed, our present study has found that the transepithelial migration of porcine IELs occurs even under physiological conditions, which exhibit age- and location-dependent characteristics. Furthermore, the integrity of the intestinal epithelial barrier (IELs resident) was evaluated by detecting tight junction protein Claudin-3. Immunohistochemistry staining showed that the intercellular movement of the IELs appeared to have no influence on the expression of the tight junctions in intestinal epithelium (Fig. R6). Therefore, we hypothesized that the porcine IELs primarily moved between the adjacent epithelial cells through the transient opening of the tight junctions, similar to what has been reported for transepithelial dendrites formation of intestinal dendritic cells.¹³. However, further studies are needed to elucidate the detailed mechanism involved. Thank you again for your kindly suggestions.

Fig.R6. Immunohistochemical analysis of intestinal tight junction. The integrity of intestinal epithelial tight junction integrity was assessed. The scale bar represents 10 μ m.

Q2. Current study uses the IPEC-J2-IEL co-culture system as a model to decipher the mechanisms invitro. However, in all of the experiments, IELs are cultured on the apical side of the IPEC-J2 cells. This is opposite to what happens invivo, where IELs are present on the basal side of the epithelium.

A2: Thank you for your careful revision. I apologize for our incorrect description of the IPEC-J2-IEL co-culture system in the “Materials and methods” section. Actually, the IPEC-J2 were seeded onto the back side of the Transwell insert membrane. Thereafter, the isolated IELs were seeded on the basolateral membrane of the IPEC-J2 and co-cultured for the indicated time. A schematic of the IPEC-J2-IEL co-culture system is shown in Fig. 3b and Fig. 4d, in which the upper face of IPEC-J2 was facing down and toward the lower compartment, and the IELs were co-cultured on the basolateral side of the IPEC-J2 cells. We have corrected our description of the culture system in the “Materials and methods” section of the revised manuscript.

Q3. Further, it was shown that upon PEDV infection, CCL2 was mainly present on the apical surface, which acts as a chemokine for the transepithelial migration of IELs. Nevertheless, in the IPEC-J2-IEL system, IELs are already on the apical side and then moving towards the basal side. How do the authors corroborate these two observations?

A3. Sorry again for our incorrect description of the IPEC-J2-IEL co-culture system in the “Materials and methods” section. In our co-culture system, the IELs were cultured on the basolateral side of the polarized cultured IPEC-J2 cells and the moved towards the apical side, which was consistent with the secretion pattern of CCL2. The corresponding corrections have been made in the revised manuscript. Thank you again for your careful review.

Q4. Next, it was found that IELs are can be directly activated by WI - PEDV. How are T cells detecting the virus? Do porcine IELS/T cells express TLRs or other pathogen recognition receptors (PRRs) detecting the virus? Similarly, how do mLN T cells are getting activated by in vitro stimulation by PEDV?

A4. Thank you very much for your careful review and questions. The explanations for your questions are as follows.

In the porcine small intestine, CD8 α^+ $\gamma\delta$ T cells and CD8 α^+ $\alpha\beta$ T cells comprised a substantial fraction of the IELs¹⁴. Among the CD8 α^+ $\alpha\beta$ T subsets, the cells expressed by CD8 $\alpha\beta^+$ heterodimer (CD8 $\alpha\beta^+$ $\alpha\beta$ T cells) were considered to be induced IELs¹⁵. These cells will likely include antigen-experienced effector or memory cells, which reside within the intestinal epithelial layer following prior infection, and recognized viral peptides presented by infected epithelial cells via their TCRs^{16, 17, 18}. Unlike induced IELs, the natural IELs (including CD8 $\alpha\alpha^+$ $\alpha\beta$ T and CD8 $\alpha\alpha^+$ $\gamma\delta$ T cells) display unusual MHC characteristics, including the ability to directly recognize nonpeptide antigens in a manner similar to antibodies.^{19, 20}

Considering that PEDV negative pigs (serum and fecal samples were negative for both PEDV antibody and nucleic acid) were used in our study, and the viral pre-activation requirement of IELs, as well as the main function of CD8 $\alpha\alpha^+$ $\alpha\beta$ IELs are widely accepted as immune regulation²¹, we speculate that the main antiviral function of the IELs were mainly undertaken by the CD8 α^+ $\gamma\delta$ T cells. Comprising 50-60% of the IEL compartment in the porcine small intestine, the $\gamma\delta$ T cells have several innate cell-like characters that allow for their early and rapid activation following recognition of cellular infection²². To accomplish these functions, the $\gamma\delta$ T cells use both the T cell receptor (TCR) and additional activating receptors (notably NKG2D, and TLR) to respond to stress-induced ligands and infection. Among these, the TLRs play critical roles in the antiviral activities of $\gamma\delta$ T cells^{23, 24}. The expression of TLRs in resting $\gamma\delta$ T cells is usually weak or undetectable but can be quickly activated and upregulated by viral infection. Based on this observation, we further tested the expression of TLR in IELs, and the IEL processing method is the same as that presented in Fig.8A. The primers used for RT-qPCR are presented in Table R1. Among the TLRs involved in viral infection, the expression of TLR2, TLR3 and TLR7 in pre-activated IELs were significantly increased after co-culturing with viral infected epithelial cells (Fig.R7, a and b). To further explore the specific TLR molecules involved in the virus recognition, the pre-activated IELs were treated with

TLR inhibitors including C29 (TLR2)²⁵, CU CPT 4a (TLR3)²⁶ and E6446 (TLR7)²⁷, which were purchased from TargetMol (USA). As shown in Fig. R7. c to e, all these inhibitor treatments had significant suppressive effects on the antiviral activity of the activated IELs compared to the control. Moreover, the inhibitors used did not affect intestinal IEL viability at their working concentration (Fig.R7, f and g). Therefore, we speculate that TLR2, TLR3 and TLR7 appear to be the key players responsible for the virus recognition by the IELs, although these results were somewhat preliminary and require further validation.

Previous studies have shown that virus could directly activate $\gamma\delta$ T cells by binding the viral receptors on the surface of the $\gamma\delta$ T cells, which enables them to acquire a pre-activated phenotype that allows for the rapid induction of effector functions following the detection of cellular infections^{28,29}. The expression of receptor molecules associated with PEDV binding^{30,31} in porcine intestinal IELs has also been confirmed by us, including APN aminopeptidase N (APN), EGFR(epidermal growth factor receptor) and transferrin receptor 1 (TfR1) (data not shown). Prior to pre-activation of WI PEDV, the porcine IELs were pre-treated with the inhibitors for these receptor molecules, including Bestin (Selleck, S1591) for APN³², Ferristatin II (Sigma-Aldrich, C1144) for TfR1³³, AG1478 (NA) (Selleck, S2728) for EGFR³⁴ (Fig. R8, a to c). At used concentrations, these inhibitors did not affect intestinal IEL viability (Fig. R8, d and e). Among them, the APN specific inhibitor Bestin significantly inhibited the antiviral function of porcine IELs in a dose-dependent manner (Fig. R8a). Therefore, the PEDV binding receptor may play an important role in mediating IEL pre-activation and the molecular mechanisms involved merit further exploration.

Resembling to intestinal IELs, the antiviral activity of MLN T cells (isolated from PBS treated pig) is also dependent upon viral pre-activation. Previous studies have shown that the $\gamma\delta$ T cells were also present in the MLN, although they only account for a small proportion of the total T cell subsets³⁵. Therefore, we speculate that the precise mechanisms of virus recognition by MLN T cells may be similar to that used by the IELs. However, the MLN T cells were actually used as a positive control in the IELs pre-activation experiment (*in vivo*), which represented the effector T cell

activated by antigen presentation. Therefore, the mechanism underlying MLN T cells activation by PEDV will be further explored in our future work.

Fig.R7. Role of Toll-like receptors (TLRs) in the antiviral function of IELs.

a-b Mock or whole inactivated porcine epidemic diarrhea virus (PEDV) pre-activated IELs were co-cultured with PEDV-infected epithelial cells for 24 h. The relative mRNA expression of TLRs (associated with viral recognition) in IELs was determined. **c-e** The pre-reactivated IELs were treated with the inhibitors for TLRs, and their antiviral activities were further detected. The cells were treated with TLR2 (C29) or TLR3 (CU CPT 4a) inhibitors for 2h, and TLR7 inhibitor for 4 h. DMSO without drugs served as a negative control. **f-g** Cell viability was determined by CCK-8 assay after-treatment of the IELs with different concentrations of TLR inhibitors. All data are the mean \pm SD,

and comparisons were performed using one-way ANOVA. $*P < 0.05$, $**P < 0.01$. The results are from at least three different experiments.

Fig.R8. Role of PEDV receptor molecule in the IELs pre-activation.

a-c The IELs were pretreated with the inhibitors for PEDV entry-related protein prior to WI PEDV pre-activation, and their antiviral activities were further detected. The cells were treated with EGFR inhibitor AG1478 for 1 h, Tfr1 inhibitor ferristatin II for 3 h, and APN inhibitor (Bestin) for 24h, respectively. DMSO without drugs served as a negative control.

f-g Cell viability was determined by CCK-8 assay after the treatment of the IELs with different concentrations of inhibitors for APN, EGFR, and Tfr1. All data are the mean \pm SD, and comparisons were performed using one-way ANOVA. $*P < 0.05$, $**P < 0.01$. The results are from at least three different experiments.

Q5. One possibility by which IELs could be activated would be contamination of sorted IELs by epithelial cells, which express various PRRs. To ensure this is not the case, can authors provide pre and post sort profiles of the IELs?

A5: Thank you very much for your suggestions. After staining with an antibody against Keratin 18 (epithelial cell marker), the profiles of the porcine intestinal IELs before and after Percoll purification were obtained (Fig. R9). FACS analysis indicated that porcine IELs used in our follow-up studies were not contaminated by epithelial cells.

Fig.R9. The profiles of the porcine IELs before and after purification.

Before or after percoll gradient centrifugation, the IELs were analyzed for purity via flow cytometry. Epithelial cells were detected by staining with cytokeratin antibodies. Flow cytometry results revealed epithelial cells contamination of less than 0.5%. Anti-epithelial cell marker PE- Keratin 18 (CK18) mAb (1:200, NBP1-97715PE) were purchased from Novus Biologicals.

Q6. In panel 5F, why are there no plaques seen in the mock of direct co-culture?

A6: I am sorry for the misunderstanding caused by the poor quality of our image. Actually, a large number of viral particles were produced in the supernatant of the

mock direct co-culture epithelial cells. We have adjusted the brightness and contrast of this image and the plaques are now visible.

Q7. In Fig 7, it was concluded that granzymes and perforin mediated antiviral activity. However, this is just a correlation as Gzms and perforins were only measured by ELISA, and no direct evidence is provided that these molecules are mediating the death of epithelial cells. Moreover, apoptosis of epithelial cells can also be caused by IFN gamma.

A7: Thank you for your insightful review. Although previous studies have shown that IELs exert their cytotoxic functions by secreting granzymes and perforin, the granzymes and perforin-mediated antiviral activity of IELs were unable to be proven by our current data. We do apologize for this over-interpreted conclusion.

To confirm the involvement of granzyme and perforin in the antiviral effects of IELs, we further conducted the following experiments. 1) The antiviral and cytotoxic activity of the IELs were determined after the granzyme and perforin inhibition treatment; 2) After treatment with IFN- γ blocking antibodies, the specific cytotoxicity of IELs against the epithelial cells was evaluated.

The antiviral activity of IEL (Fig. R10a) and the IEL-mediated epithelial cells apoptosis (Fig. R10b) were not influenced by treating with the perforin inhibitor concanamycin A (CMA) or granzyme inhibitor 3,4 Dichloroisocoumarin (DCI)^{36,37}. It has also been confirmed that at the concentrations used, the two inhibitors will not affect the viability of the IELs (Fig. R10d). However, the IELs-induced apoptosis of epithelial cells was significantly inhibited by the IFN- γ antibody (25 ug) treatment (Fig. R10c). The abovementioned results are consistent with the results in Fig.7H, where the antiviral effect of IELs could be almost entirely antagonized by IFN- γ antibody (Fig.7H).

Therefore, although high levels of perforin were detected in the co-culture system, we believed that the antiviral function of the activated IELs was mainly mediated by IFN- γ secretion that induced robust antiviral response in the virus-infected cells. We have added these supplementary results in the "Results" section of the revised

manuscript. The relevant text in both the "Discussion" and "Conclusion" sections were also corrected. Thank you again for your careful review.

Fig.R10. The antiviral activity of IELs was independent of their cytotoxic effects.

a, b The pre-reactivated IELs were treated with CMA at two concentrations (1 and 0.5 μM) or DCI at two concentrations (20 and 50 μM) for 3 h, and their antiviral activities were further detected by co-culturing with virus-infected epithelial cells. Both of the inhibitors were also maintained for the whole duration of the co-culturing process. The intracellular viral RNA expression (a) and extracellular LDH activity (b) in each experimental group were determined. **c** Mock or whole inactivated virus (WIV) porcine epidemic diarrhea virus (PEDV) pre-activated IELs were co-cultured with PEDV-infected epithelial cells for 24 h. During co-cultivation, blocking antibodies against IFN-γ were added to the medium with a certain concentration gradient. The extracellular LDH activities in each experimental group were determined. **d** Cell viability was determined by CCK-8 assay after the treatment of the IELs with different concentrations of inhibitors for perforin and granzyme B. All data are the mean ± SD and comparisons were performed using one-way ANOVA. * $P < 0.05$, ** $P < 0.01$. The

results are from at least three different experiments.

Q8. In conclusion, the authors found a transepithelial behavior of porcine IELs, but its implications are not very clear from the current data presented in the manuscript.

Therefore, the paper's title seems to be an overstatement in concluding that transepithelial migration leads to the antiviral state.

A8. Thank you very much for your careful review and kind reminder. We apologize for over-interpreting our research conclusion. We acknowledge that the direct relationship between the observed migratory behavior of IELs as well as their antiviral activity remains speculative based on the data we currently have and requires further validation. We have toned down these related conclusions and amended the title in the revised manuscript.

Table S1. Primer sequences used for RT-qPCR

Genes	Primers	Sequence (5'-3')*
TLR2	Forward	GAGTCTGCCACAACACTCAAAGA
	Reverse	CAGAACTGACAACATGGGTAGAA
TLR3	Forward	GAGCAGGAGTTTGCCTTGTC
	Reverse	GGAGGTCATCGGGTATTTGA
TLR4	Forward	TCATCCAGGAAGGTTTCCAC
	Reverse	TGTCTCCCACTCCAGGTAG
TLR7	Forward	TCTGCCCTGTGATGTCAGTC
	Reverse	GCTGGTTTCCATCCAGGTAA
TLR8	Forward	CTGGGATGCTTGGTTCATCT
	Reverse	CATGAGGTTGTCGATGATGG
TLR9	Forward	AGGGAGACCTCTATCTCCGC
	Reverse	AAGTCCAGGGTTTCCAGCTT
CCR2	Forward	ATGCCAGTTTTCTACGGGG
	Reverse	CCGGGCACTTGCTTTAGAGA
CCR5	Forward	TGGTCAGAGGAGCTGAGACA
	Reverse	AGAAGGGACTCGTCGTTTGA

References

1. Ho WZ, *et al.* Centrifugal enhancement of human immunodeficiency virus type 1 infection and human cytomegalovirus gene expression in human primary monocyte/macrophages in vitro. *Journal of leukocyte biology* **53**, 208-212 (1993).
2. Guo J, Wang W, Yu D, Wu Y. Spinoculation triggers dynamic actin and cofilin activity that facilitates HIV-1 infection of transformed and resting CD4 T cells. *J Virol* **85**, 9824-9833 (2011).
3. O'Doherty U, Swiggard WJ, Malim MH. Human immunodeficiency virus type 1 spinoculation enhances infection through virus binding. *J Virol* **74**, 10074-10080 (2000).
4. Roberts AI, *et al.* NKG2D receptors induced by IL-15 costimulate CD28-negative effector CTL in the tissue microenvironment. *J Immunol* **167**, 5527-5530 (2001).
5. Vandereyken M, James OJ, Swamy M. Mechanisms of activation of innate-like intraepithelial T lymphocytes. *Mucosal Immunol* **13**, 721-731 (2020).
6. Cheroutre H, Lambolez F, Mucida D. The light and dark sides of intestinal intraepithelial lymphocytes. *Nat Rev Immunol* **11**, 445-456 (2011).
7. Mikulak J, *et al.* NKp46-expressing human gut-resident intraepithelial V δ 1 T cell subpopulation exhibits high antitumor activity against colorectal cancer. *JCI Insight* **4**, (2019).
8. Wang H-C, Klein JR. Multiple levels of activation of murine CD8+ intraepithelial lymphocytes defined by OX40 (CD134) expression: effects on cell-mediated cytotoxicity, IFN- γ , and IL-10 regulation. *The Journal of Immunology* **167**, 6717-6723 (2001).
9. Croft M, So T, Duan W, Soroosh P. The significance of OX40 and OX40L to T-cell biology and immune disease. *Immunological reviews* **229**, 173-191 (2009).
10. Mabbott NA, Donaldson DS, Ohno H, Williams IR, Mahajan A. Microfold (M) cells: important immunosurveillance posts in the intestinal epithelium. *Mucosal immunology* **6**, 666-677 (2013).
11. Niess JH, *et al.* CX3CR1-mediated dendritic cell access to the intestinal lumen and bacterial clearance. *Science* **307**, 254-258 (2005).
12. Müller S, Bühler-Jungo M, Mueller C. Intestinal intraepithelial lymphocytes exert potent protective cytotoxic activity during an acute virus infection. *The Journal of Immunology* **164**, 1986-1994 (2000).

13. Rescigno M, *et al.* Dendritic cells express tight junction proteins and penetrate gut epithelial monolayers to sample bacteria. *Nature immunology* **2**, 361-367 (2001).
14. Wiarda JE, Trachsel JM, Bond ZF, Byrne KA, Gabler NK, Loving CL. Intraepithelial T cells diverge by intestinal location as pigs age. *Frontiers in immunology* **11**, 1139 (2020).
15. Hayday A, Theodoridis E, Ramsburg E, Shires J. Intraepithelial lymphocytes: exploring the Third Way in immunology. *Nature immunology* **2**, 997-1003 (2001).
16. Gebhardt T, Wakim LM, Eidsmo L, Reading PC, Heath WR, Carbone FR. Memory T cells in nonlymphoid tissue that provide enhanced local immunity during infection with herpes simplex virus. *Nat Immunol* **10**, 524-530 (2009).
17. Ariotti S, *et al.* Tissue-resident memory CD8+ T cells continuously patrol skin epithelia to quickly recognize local antigen. *Proc Natl Acad Sci U S A* **109**, 19739-19744 (2012).
18. van de Sandt CE, *et al.* Challenging immunodominance of influenza-specific CD8(+) T cell responses restricted by the risk-associated HLA-A*68:01 allomorph. *Nat Commun* **10**, 5579 (2019).
19. Cheroutre H, Lambolez F, Mucida D. The light and dark sides of intestinal intraepithelial lymphocytes. *Nature Reviews Immunology* **11**, 445-456 (2011).
20. Born WK, Aydinoglu MK, O'Brien RL. Diversity of $\gamma\delta$ T-cell antigens. *Cellular & molecular immunology* **10**, 13-20 (2013).
21. McDonald BD, Jabri B, Bendelac A. Diverse developmental pathways of intestinal intraepithelial lymphocytes. *Nature Reviews Immunology* **18**, 514-525 (2018).
22. Bonneville M, O'Brien RL, Born WK. $\gamma\delta$ T cell effector functions: a blend of innate programming and acquired plasticity. *Nature Reviews Immunology* **10**, 467-478 (2010).
23. Dar AA, Patil RS, Chiplunkar SV. Insights into the relationship between toll like receptors and gamma delta T cell responses. *Frontiers in immunology* **5**, 366 (2014).
24. Wesch D, Peters C, Oberg H-H, Pietschmann K, Kabelitz D. Modulation of $\gamma\delta$ T cell responses by TLR ligands. *Cellular and molecular life sciences* **68**, 2357-2370 (2011).
25. Mistry P, *et al.* Inhibition of TLR2 signaling by small molecule inhibitors targeting a pocket within the TLR2 TIR domain. *Proc Natl Acad Sci U S A* **112**, 5455-5460 (2015).
26. Cheng K, Wang X, Yin H. Small-molecule inhibitors of the TLR3/dsRNA complex. *Journal of the American Chemical Society* **133**, 3764-3767 (2011).

27. Lamphier M, *et al.* Novel small molecule inhibitors of TLR7 and TLR9: mechanism of action and efficacy in vivo. *Mol Pharmacol* **85**, 429-440 (2014).
28. Dong P, *et al.* $\gamma\delta$ T cells provide protective function in highly pathogenic avian H5N1 influenza A virus infection. *Frontiers in immunology* **9**, 2812 (2018).
29. Lu Y, *et al.* The interaction of influenza H5N1 viral hemagglutinin with sialic acid receptors leads to the activation of human $\gamma\delta$ T cells. *Cellular & molecular immunology* **10**, 463-470 (2013).
30. Wang Q, Vlasova AN, Kenney SP, Saif LJ. Emerging and re-emerging coronaviruses in pigs. *Current opinion in virology* **34**, 39-49 (2019).
31. Yang L, *et al.* Porcine epidemic diarrhea virus-induced epidermal growth factor receptor activation impairs the antiviral activity of type I interferon. *Journal of virology* **92**, e02095-02017 (2018).
32. Cui SX, *et al.* Targeting aminopeptidase N (APN/CD13) with cyclic-imide peptidomimetics derivative CIP-13F inhibits the growth of human ovarian carcinoma cells. *Cancer Lett* **292**, 153-162 (2010).
33. Byrne SL, *et al.* Ferristatin II promotes degradation of transferrin receptor-1 in vitro and in vivo. *PLoS One* **8**, e70199 (2013).
34. Pogrmic-Majkic K, Samardzija Nenadov D, Fa S, Stanic B, Trninic Pjevic A, Andric N. BPA activates EGFR and ERK1/2 through PPAR γ to increase expression of steroidogenic acute regulatory protein in human cumulus granulosa cells. *Chemosphere* **229**, 60-67 (2019).
35. Krishna VD, *et al.* Immune responses to porcine epidemic diarrhea virus (PEDV) in swine and protection against subsequent infection. *PLoS One* **15**, e0231723 (2020).
36. Boivin WA, *et al.* Granzyme B cleaves decorin, biglycan and soluble betaglycan, releasing active transforming growth factor- β 1. *PLoS One* **7**, e33163 (2012).
37. Sukeda M, Shiota K, Kondo M, Nagasawa T, Nakao M, Somamoto T. Innate cell-mediated cytotoxicity of CD8(+) T cells against the protozoan parasite *Ichthyophthirius multifiliis* in the ginbuna crucian carp, *Carassius auratus langsdorfii*. *Dev Comp Immunol* **115**, 103886 (2021).

REVIEWERS' COMMENTS:

Reviewer #1 (Remarks to the Author):

The authors have responded satisfactorily to my concerns and the manuscript is improved.

However I believe there is an error in the y-axis legend for Figures 1f, 2b and 2e. I understand that the % of cells is now included underneath the graphs. However it is unclear what the main part of the graphs show - they often sum to above 100, so is this absolute number? The axis titles say proportion (%) but this cannot be correct.

Reviewer #2 (Remarks to the Author):

The authors have done a commendable job answering and clarifying the comments. I highly recommend this work for publication as this will be of great interest to the innate lymphoid cell and IEL community.

COMMSBIO-21-1856B

Title: Porcine intraepithelial lymphocytes undergo migration and produce an antiviral response following intestinal virus infection

Point to point response

Reviewers' comments:

Reviewer #1 (Remarks to the Author):

The authors have responded satisfactorily to my concerns and the manuscript is improved. However, I believe there is an error in the y-axis legend for Figures 1f, 2b and 2e. I understand that the % of cells is now included underneath the graphs. However, it is unclear what the main part of the graphs show - they often sum to above 100, so is this absolute number? The axis titles say proportion (%) but this cannot be correct.

A: Thank you very much for your approval and opinions. I apologize for this mistake. Actually, the y axis indicated the absolute number of T cells and the proportion (%) of T cells was included underneath the graphs. We have modified the figures, including Fig. 1f, Fig. 2b and Fig. 2e.

Reviewer #2 (Remarks to the Author):

The authors have done a commendable job answering and clarifying the comments. I highly recommend this work for publication as this will be of great interest to the innate lymphoid cell and IEL community.

A: Thank you very much for your help and approval.